# Alternative splicing downstream of EMT enhances phenotypic plasticity and malignant behavior in colon cancer

Tong Xu[1], Mathijs Verhagen[1], Rosalie Joosten[1], Wenjie Sun[2], Andrea Sacchetti[1], Leonel Munoz Sagredo[3,4], Véronique Orian-Rousseau[3], Riccardo Fodde[1]*

[1]Department of Pathology, Erasmus University Medical Center, Rotterdam, Netherlands; [2]Laboratory of Genetics and Developmental Biology, Institute Curie, Paris, France; [3]Institute of Biological and Chemical Systems - Functional Molecular Systems (IBCS FMS), Karlsruhe Institute of Technology, Karlsruhe, Germany; [4]Faculty of Medicine, University of Valparaiso, Valparaiso, Chile

**Abstract** Phenotypic plasticity allows carcinoma cells to transiently acquire the quasi-mesenchymal features necessary to detach from the primary mass and proceed along the invasion-metastasis cascade. A broad spectrum of epigenetic mechanisms is likely to cause the epithelial-to-mesenchymal (EMT) and mesenchymal-to-epithelial (MET) transitions necessary to allow local dissemination and distant metastasis. Here, we report on the role played by alternative splicing (AS) in eliciting phenotypic plasticity in epithelial malignancies with focus on colon cancer. By taking advantage of the coexistence of subpopulations of fully epithelial (EpCAM$^{hi}$) and quasi-mesenchymal and highly metastatic (EpCAM$^{lo}$) cells in conventional human cancer cell lines, we here show that the differential expression of *ESRP1* and other RNA-binding proteins (RBPs) downstream of the EMT master regulator *ZEB1* alters the AS pattern of a broad spectrum of targets including *CD44* and *NUMB*, thus resulting in the generation of specific isoforms functionally associated with increased invasion and metastasis. Additional functional and clinical validation studies indicate that both the newly identified RBPs and the CD44s and NUMB2/4 splicing isoforms promote local invasion and distant metastasis and are associated with poor survival in colon cancer. The systematic elucidation of the spectrum of EMT-related RBPs and AS targets in epithelial cancers, apart from the insights in the mechanisms underlying phenotypic plasticity, will lead to the identification of novel and tumor-specific therapeutic targets.

*For correspondence:
r.fodde@erasmusmc.nl

## Editor's evaluation

This fundamental study provides a valuable analysis of the splicing landscape in colon cancer cells that have properties intermediate between those typically found in primary cancers ("epithelial") and those that are spreading by metastasis ("mesenchymal"). The strength of evidence provided is solid and convincing and supports current ideas that changes in the way that RNA from particular genes is processed plays a key role in cancer spread.

## Introduction

Colon cancer still represents one of the major causes of cancer-related morbidity and mortality worldwide. Apart from its high incidence, the adenoma-carcinoma sequence along which colon cancer progresses has served as a classic model to elucidate the underlying genetic alterations representative of virtually all of the hallmarks of cancers (*Hanahan, 2022*), possibly with the only exception of

'*activating invasion and metastasis (unlocking phenotypic plasticity; non-mutational epigenetic reprogramming)*'. As also reported in other epithelial cancers, the several steps of the invasion-metastasis cascade are not caused by genetic alterations but rather by transient morphological and gene expression changes of epigenetic nature (*Bernards and Weinberg, 2002*; *Reiter et al., 2018*). In this context, epithelial-mesenchymal transition (EMT) and its reverse mesenchymal-epithelial transition (MET) likely represent the main mechanisms underlying local dissemination and distant metastasis (*Thiery et al., 2009*; *Brabletz et al., 2005*). EMT is triggered at the invasive front of the primary colon carcinoma in cells earmarked by nuclear β-catenin and enhanced Wnt signaling, as the result of their physical and paracrine interactions with the microenvironment (*Fodde and Brabletz, 2007*). The acquisition of quasi-mesenchymal features allows local invasion and dissemination through the surrounding stromal compartment. Of note, EMT/MET should not be regarded as binary processes in view of the existence of metastable hybrid E/M states (partial EMT [pEMT]) endowed with phenotypic plasticity and likely to underlie the reversible morphological and functional transitions necessary to successfully complete the invasion-metastasis cascade (*Teeuwssen and Fodde, 2019*).

The molecular basis of the epigenetic changes underlying EMT and MET is likely to encompass a broad spectrum of mechanisms ranging from chromatin remodeling and histone modifications to promoter DNA methylation, non-coding RNAs (e.g. microRNAs), and alternative splicing (AS). The inclusion/exclusion of specific exons in mature mRNAs results in different protein isoforms with distinct biological functions. AS occurs in 92–94% of human genes leading to enriched protein density (*Wang et al., 2008*; *Blencowe, 2006*). Several sequence-specific RNA-binding proteins (RBPs) have been identified which bind pre-mRNAs to control AS in context-dependent fashion (*Fu and Ares, 2014*). Multiple cancer-specific AS variants have been found to underlie progression and metastasis (*Kahles et al., 2018*). Likewise, AS has been suggested to play key roles in EMT/MET (*Roy Burman et al., 2021*; *Oltean and Bates, 2014*) and phenotypic plasticity (*Biamonti et al., 2019*) in cancer by expression changes in RBP-encoding genes and their consequences for the modulation of downstream AS targets.

The *ESRP1* (epithelial splicing regulatory protein 1) gene encodes for an epithelial-specific RBP and splicing regulator shown to play a central role in EMT by modulating AS of EMT-associated genes including *FGFR2*, Mena, *CD44,* and p120-catenin (*Thiery et al., 2009*). Relevant to the present study, ESRP1 was reported to regulate the EMT from CD44v (variable) to CD44s (standard) isoforms in breast and lung cancer progression (*Brown et al., 2011*; *Yae et al., 2012*). As for colon cancer, whether ESRP1 regulates AS of CD44 and other target genes downstream of EMT/MET activation during invasion and metastasis is yet poorly understood.

Recently, we identified and thoroughly characterized subpopulations of CD44hi/EpCAMlo cells (here referred to as EpCAMlo) that coexist within immortalized colon cancer cell lines with their epithelial counterparts (CD44hi/EpCAMhi; for brevity EpCAMhi) through stochastic state transitions governed by phenotypic plasticity and pEMT (*Sacchetti et al., 2021*). Accordingly, EpCAMlo cells feature highly invasive and metastatic capacities. Here, we took advantage of these in vitro models of phenotypic plasticity to test the hypothesis according to which AS driven by upstream RBPs underlie EMT (and MET). Among the identified AS targets, specific CD44 and NUMB isoforms were shown to play specific and unexpected roles in stemness and cancer. Moreover, we provide an extensive list of additional EMT-related RBPs and AS targets and show that many are conserved in other epithelial malignancies. Likewise, RBPs and AS targets differentially expressed among distinct carcinoma types are likely to reflect the distinct modalities through which these malignant cells metastasize.

## Results

### Differential expression of RBPs in the quasi-mesenchymal and highly metastatic EpCAMlo colon cancer cells affects AS of a broad spectrum of downstream target genes

As previously reported, the EpCAMlo subpopulation of colon cancer cells is earmarked by increased expression of the *ZEB1* transcription factor, responsible for EMT activation and for their quasi-mesenchymal and highly metastatic phenotype (*Sacchetti et al., 2021*). It has been established that in breast and pancreatic cancer *ZEB1*-driven EMT downregulates the expression of the RBP and splicing regulator *ESRP1* as part of a self-enforcing feedback loop (*Preca et al., 2015*). Accordingly,

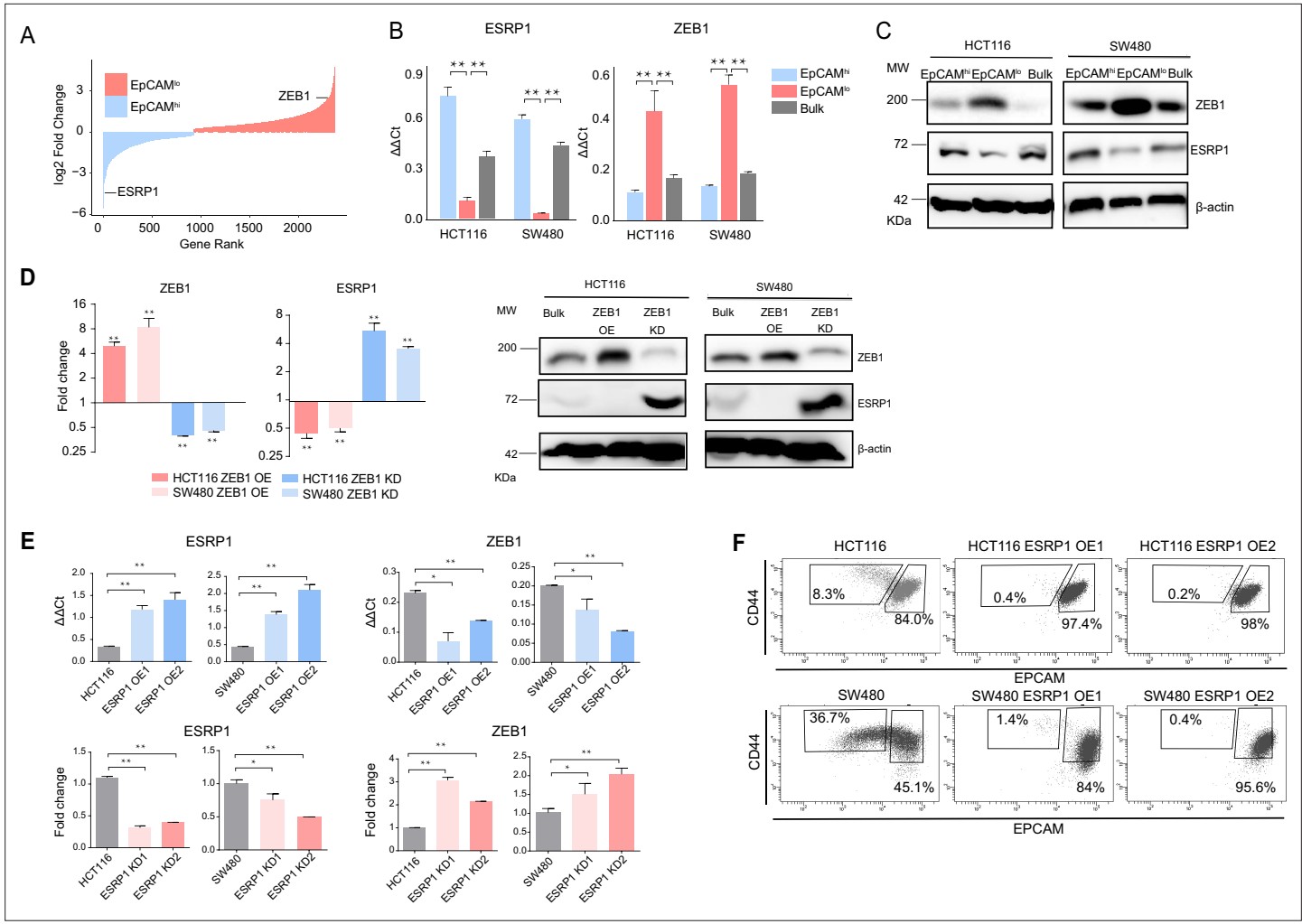

**Figure 1.** *ZEB1* and *ESRP1* differential expression in quasi-mesenchymal and highly metastatic EpCAM[lo] colon cancer cells. (**A**) Gene rank plot showing differentially expressed genes between EpCAM[hi] and EpCAM[lo] with combined analysis of HCT116 and SW480. (**B**) RT-qPCR *ESRP1* and *ZEB1* expression analysis of HCT116 and SW480 EpCAM[hi], EpCAM[lo], and bulk subpopulations. *GAPDH* expression was used as control (means ± SEM, n=3). **=p < 0.01. (**C**) ESRP1 and ZEB1 western analysis in HCT116 and SW480 EpCAM[hi], EpCAM[lo], and bulk fractions. β-Actin was used as loading control. (**D**) RT-qPCR and western analysis of *ZEB1* and *ESRP1* expression in *ZEB1*-OE and -KD HCT116 and SW480 cells. Expression values were normalized in each sample with those from the parental HCT116 and SW480 cell lines. HCT116 and SW480 cells transduced with the sh*ZEB1* lentivirus were induced by 1 μg/mL doxycycline for 72 hr. Expression values were normalized with those from non-induced cells; *GAPDH* expression was employed as control (means ± SEM, n=3). *=p < 0.05, **=p < 0.01. β-Actin was used as loading control. (**E**) RT-qPCR *ZEB1* and *ESRP1* expression analysis in *ESRP1*-OE and -KD HCT116 and SW480 cells. Two independent *ESRP1*-OE clones were selected for each cell line. Expression values were normalized in each sample with those from the parental HCT116 and SW480 cell lines. HCT116 and SW480 cells transduced with the sh*ESRP1* lentivirus were induced by 1 μg/mL doxycycline for 72 hr. Two independent clones were selected for each cell line. Expression values were normalized with those from non-induced cells; *GAPDH* expression was employed as control (means ± SEM, n=3). *=p < 0.05, **=p < 0.01. (**F**) CD44/EpCAM FACS analysis of HCT116 and SW480 EpCAM[lo] and EpCAM[hi] subpopulations in ESRP1-OE cells. Two independent clones are shown for each cell lines.

The online version of this article includes the following source data and figure supplement(s) for figure 1:

**Source data 1.** Original files and labelled bands of western blots in *Figure 1C–D*.

**Figure supplement 1.** *ESRP1* and RNA-binding proteins (RBPs) functional and expression analysis in cell lines and patient-derived colon cancers.

among the top differentially expressed genes (DEGs) between EpCAM[lo] and EpCAM[hi] in SW480 and HCT116 colon cancer cells, *ESRP1* was found to be downregulated both at the RNA and at the protein level in the quasi-mesenchymal subpopulation where *ZEB1* expression is upregulated (*Figure 1A–C*). Gain- and loss-of-function analyses of both genes confirmed the inter-dependence of their expression levels in both cell lines (*Figure 1D–E*). Of note, *ESRP1* overexpression in the HCT116 and SW480 cell lines resulted in the dramatic reduction of their EpCAM[lo] subpopulations and the expansion of the

epithelial bulk (EpCAM^hi), as shown by FACS analysis (*Figure 1F*, *Figure 1—figure supplement 1A*). However, *ESRP1* knockdown (KD) gave rise to less clear and extremely variable results among the individual clones analyzed by FACS, in particular in the SW480 cell line. More coherent and representative results were obtained with the pools of the KD transfections (*Figure 1—figure supplement 1B*).

These results suggest that RBPs other than ESRP1 are likely to be involved in the AS regulation of the EpCAM^lo colon cancer subpopulation. Indeed, by taking advantage of the RBPDB database (*Cook et al., 2011*), we found that, apart from *ESRP1*, consistent differential expression in the quasi-mesenchymal subpopulation of both cell lines was observed for *ESRP2*, *RBM47*, *MBNL3* (downregulated) and *NOVA2*, *MBNL2* (upregulated). Other RBPs were found to be differentially expressed though in only one of the two cell lines (*Figure 1—figure supplement 1C*). In validation of the clinical relevance of the RBPs found to be differentially expressed between the EpCAM^hi/lo subpopulations derived from the SW480 and HCT116 cell lines, the RBP-coding genes *QKI*, *RBM24*, and *MBNL2* (up in EpCAM^lo), and *ESRP1/2* and *RBM47* (down in EpCAM^lo) were found to be respectively up- and down-regulated in the consensus molecular subtype 4 (CMS4) of colon cancers, responsible for ~25% of the cases and earmarked by poor prognosis and a pronounced mesenchymal component (*Figure 1—figure supplement 1D*; *Guinney et al., 2015*).

Differentially spliced target genes between EpCAM^lo and EpCAM^hi colon cancer cells from the SW480 and HCT116 cell lines were selected based on exon skip splicing events with ΔPSI (differential percentage spliced in) values >10%. The PSI value ranges from 0 to 1 and is a measurement of the percentage of isoform with an alternative exon included (*Schafer et al., 2015*). This resulted in a large and rather heterogeneous group of alternative spliced targets (n=1495; *Supplementary file 1a*) with no clear enrichment in any specific gene ontology class (data not shown). In order to identify differentially spliced target genes in RBP-specific fashion, we took advantage of RNAseq data sets from previous *ESRP1-*, *ESRP2-*, *RBM47-*, and *QKI*-KD studies in different cancer cell lines and compared them with our own AS data relative to the EpCAM^hi/lo colon cancer subpopulations (*Sacchetti et al., 2021*; *Figure 2A* and *Figure 2—figure supplement 1*). A total of 32 common skipped exons events in 20 genes were identified between EpCAM^lo colon (both cell lines) and *ESRP1* KD H358 lung cancer cells (*Yang et al., 2016*; *Figure 2A*). More extensive lists of common *ESRP1* AS events and target genes were obtained when the SW480 and HCT116 cell lines were individually compared with the lung cancer study (*Supplementary file 1B-C*). As for the AS targets of RBPs other than *ESRP1*, based on the available RNAseq data from KD studies of *ESRP2* (in the LNCaP cell line *Nieto et al., 2016*), *RBM47* (H358 *Yang et al., 2016*), and *QKI* (CAL27; GEO Accession: GSM4677985), several common and unique genes were found (*Figure 2—figure supplement 1* and *Supplementary file 2*). Notably, four EMT-related genes (*CTNND1 Hernández-Martínez et al., 2019*, LSR *Shimada et al., 2021*, *SLK Conway et al., 2017*, and *TCF7L2 Karve et al., 2020*) were common to all RBP KD studies analyzed (*Figure 2—figure supplement 1*).

## The CD44s and NUMB2/4 ESRP1-specific AS isoforms are preferentially expressed in EpCAM^lo colon cancer cells

From the newly generated lists of RBP-specific AS targets, we selected *CD44* and *NUMB* for further analysis, based both on their *ESRP1*-specific AS patterns and on their well-established roles in EMT, stemness/differentiation, and cancer progression.

CD44, a transmembrane cell surface glycoprotein, has been show to play key roles in inflammatory responses and in cancer metastasis (*Orian-Rousseau, 2015*). The *CD44* gene encompasses 20 exons of which 1–5 and 16–20 are constant and exist in all isoforms. In contrast, exons 6–14, also referred to as variants exons v2-v10, are alternatively spliced and often deregulated in cancer (*Orian-Rousseau, 2015*). The *NUMB* gene and its protein product have been involved in a broad spectrum of cellular phenotypes including cell fate decisions, maintenance of stem cell niches, asymmetric cell division, cell polarity, adhesion, and migration. In cancer, NUMB is a tumor suppressor that regulates, among others, Notch and Hedgehog signaling (*Pece et al., 2011*). The mammalian *NUMB* gene encodes for four isoforms, ranging from 65 to 72 KD, differentially encompassing two key functional domains, that is, the amino-terminal phosphotyrosine-binding domain, and a C-terminal proline-rich region domain (*Pece et al., 2011*).

Based on the above ΔPSI-based AS analysis, decreased expression of CD44v (variable) isoforms was observed in EpCAM^lo and *ESRP1*-KD cells, accompanied by increased CD44s (standard) isoform

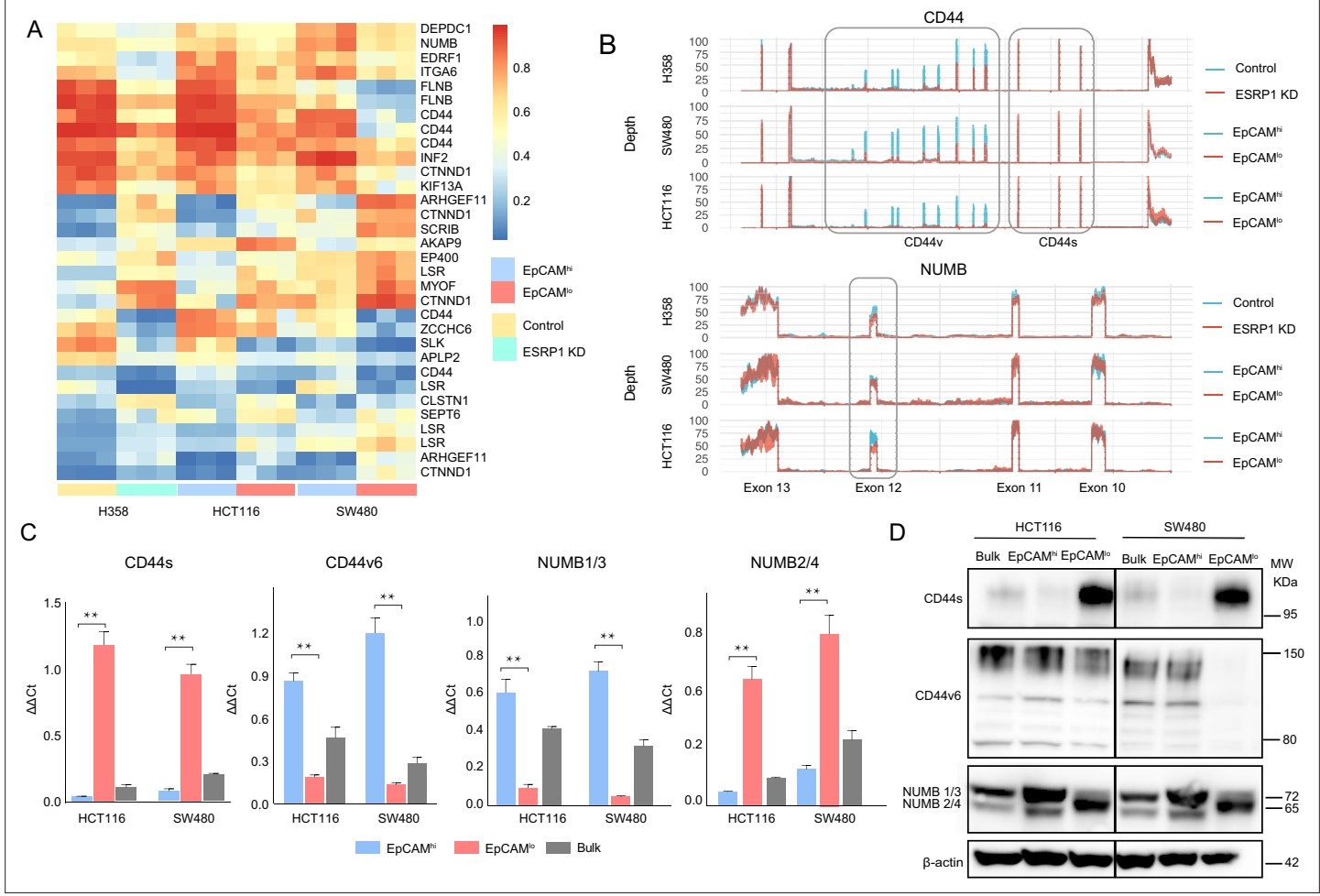

**Figure 2.** *ESRP1* downregulation in EpCAM^lo^ colon cancer cells affects alternative splicing (AS) of *CD44* and *NUMB* among a broad spectrum of downstream target genes. (**A**) Heatmap of common AS events between RNAseq data from a previous *ESRP1*-KD study in human non-small cell lung cancer cells (H358) (*Yang et al., 2016*) and our own HCT116 and SW480 EpCAM^hi^ and EpCAM^lo^ RNAseq data (*Sacchetti et al., 2021*). The gene list on the right of the heatmap encompasses AS variants earmarked by ΔPSI (differential percentage spliced in) > 0.1. (**B**) *CD44* and *NUMB* exon peak plots relative to the AS analysis of the RNAseq data obtained from a previous *ESRP1*-KD study in human non-small cell lung cancer cells (H358; upper graph) (*Yang et al., 2016*) and from our own HCT116 (middle graph) and SW480 (lower graph) EpCAM^hi/lo^ analysis (*Sacchetti et al., 2021*). Each peak plot depicts the expression of specific exons; the height of each peak is indicative of the expression level of the specific exons. CD44v: CD44 exons v2 to v10. CD44v and CD44s, and NUMB exon 12 is highlighted by gray rectangles. (**C**) RT-qPCR expression analysis of *CD44*s, *CD44v6*, *NUMB1/3*, and *NUMB2/4* isoforms in HCT116 and SW480 EpCAM^hi^, EpCAM^lo^, and bulk subpopulations. Expression of the constitutive *CD44* and *NUMB* exons was employed to normalize the results (means ± SEM, n=3). **=p < 0.01. (**D**) Western analysis of CD44s, CD44v6, and NUMB isoforms in HCT116 and SW480 EpCAM^hi^, EpCAM^lo^, and bulk subpopulations. Please note that the molecular weight of CD44v6 is expected to range between 80 and 150 kDa (*Azevedo et al., 2018*, *Ponta et al., 2003*). β-Actin was used as loading control.

The online version of this article includes the following source data and figure supplement(s) for figure 2:

**Source data 1.** Original files and labelled bands of western blots in **Figure 2D**.

**Figure supplement 1.** *ESRP1/2*-, *RBM47*-, and *QKI*-regulated alternative splicing (AS) targets.

expression (**Figure 2B**). Likewise, the NUMB2/4 isoforms appear to be preferentially expressed in EpCAM^lo^ and *ESRP1*-KD, accompanied by decreased NUMB1/3 expression (**Figure 2B**, **Figure 2— figure supplement 1B**). RT-qPCR and western analyses validated these in silico data: CD44s and NUMB2/4 isoforms were preferentially expressed in EpCAM^lo^ colon cancer cells, in contrast with the increased CD44v and NUMB1/3 levels in EpCAM^hi^ cells (**Figure 2C–D**). In view of its previously suggested role in invasion and metastasis (*Todaro et al., 2014*), we focused on the CD44v6 isoform.

As reported above, AS events at the *NUMB* and *CD44* genes correlate with decreased ESRP1 expression. To confirm this observation, we up- and downregulated *ESRP1* in the SW480 and HCT116

cell lines. The dox-inducible shRNA vector used for the KD studies reduces ESRP1 expression by 5- to 10-fold (*Figure 1D–E*) and resulted in the upregulation of the CD44s and NUMB2/4 isoforms at the mRNA and protein level in both cell lines (*Figure 3A–B* and *Figure 3—figure supplement 1A–B*). Likewise, *ESRP1* overexpression led to an increase in the CD44v6 and NUMB1/3 isoforms, found in association with the bulk of epithelial colon cancer cells (*Figure 3C–D* and *Figure 3—figure supplement 1C–D*).

## Transcriptional and functional consequences of the CD44s and NUMB2/4 isoforms on colon cancer invasion and metastasis

In order to elucidate the functional contribution exerted by the newly identified CD44s and NUMB2/4 isoforms on the overall invasive and metastatic capacities of colon cancer cells, we first ectopically expressed each of them (individually and in combination for NUMB1/3 and 2/4) in the HCT116 and SW480 cell lines (*Figure 3—figure supplement 1E–H*), and analyzed their consequences in vitro by cell proliferation, transwell migration assay, RT-qPCR, western, FACS, and RNAseq, and in vivo by spleen transplantation. A significant increase in migratory capacity (*Figure 3—figure supplement 2A–B*), comparable to that of EpCAM$^{lo}$ cells sorted from the parental lines, was observed in SW480 and HCT116 upon overexpression of the CD44s and NUMB2/4 isoforms (*Figure 3—figure supplement 2A–B*). Likewise, ectopic expression of the single NUMB2 or -4 isoforms resulted in increased migration rates when compared with NUMB1 and -3. In contrast, overexpression of CD44v6 and NUMB1/3, normally prevalent in the epithelial bulk (EpCAM$^{hi}$) of both cell lines, did not affect their migratory properties (*Figure 3—figure supplement 2A–B*).

In agreement with the migration assays, overexpression of CD44s and NUMB2/4 results in the significant upregulation of the EMT transcription factors (EMT-TFs) *ZEB1*, accompanied by the up- and downregulation regulation of mesenchymal and epithelial markers such as *VIM* (vimentin), *CDH1* (E-cadherin), and *EpCAM*, respectively (*Figure 3—figure supplement 2C*). Of note, expression of *ESRP1*, the main upstream splicing regulator of both CD44 and NUMB, was also decreased in CD44s- and NUMB2/4-OE cells, in confirmation of the self-enforcing feedback loop that characterize its interaction with ZEB1 and EMT activation (*Preca et al., 2015*). In agreement with the well-established regulation of Notch signaling by NUMB isoforms (*Pece et al., 2011*), established Notch target genes and were accordingly up- (*HES1*, *HEY1*) and downregulated (*ID2*) upon overexpression of NUMB2/4 (*Figure 3—figure supplement 2D*).

FACS analysis was then employed to evaluate the overall effect of the ectopic expression of the specific CD44 and NUMB isoforms on the relative percentages of the EpCAM$^{hi/lo}$ subpopulations in the HCT116 and SW480 cell lines. As shown in *Figure 4A*, CD44s overexpression led to a dramatic increase of the EpCAM$^{lo}$ subpopulation at the expenses of EpCAM$^{hi}$ cells. The opposite effect was observed with CD44v6, that is, the enlargement of the EpCAM$^{hi}$ gate and the corresponding decrease of EpCAM$^{lo}$ cells. As for NUMB, ectopic expression of NUMB2/4 significantly increased the relative proportion of EpCAM$^{lo}$ cells while reducing the size of the EpCAM$^{hi}$ subpopulation, while the opposite was observed with NUMB1/3 (*Figure 4B–C*). Of note, the single NUMB2 and NUMB4 isoforms appear dominant in their capacity to enlarge the HCT116 and SW480 EpCAM$^{lo}$ subpopulations, respectively. The same was true for NUMB1 and NUMB3 in the consequences of their ectopic expression in reducing the size of the HCT116 and SW480 EpCAM$^{lo}$ fractions, respectively (*Figure 4B–C*). In agreement with the RTqPCR analysis of EMT markers, CD44s overexpression negatively affected overall proliferation rates in both cell lines, whereas the opposite was observed upon CD44v6 expression (*Figure 4—figure supplement 1A–B*). Likewise, NUMB1/3 expression positively affected proliferation rates in HCT116 and SW480, whereas the NUMB2/4 isoforms exert the opposite effects. In both cases, synergistic effects were observed upon co-expression of NUMB1/3 and 2/4, when compared to the individual isoforms (*Figure 4—figure supplement 1C–D*).

In order to assess the in vivo consequences of the ectopic expression of the CD44 and NUMB isoforms on the capacity of colon cancer cells to form metastatic lesions in the liver, parental HCT116 and SW480 cells and their CD44s-, CD44v6-, NUMB1/3-, and NUMB1/4-overexpressing (OE) counterparts were injected in the spleen of immune-incompetent recipient mice. In agreement with the in vitro results, overexpression of both NUMB2/4 and CD44s isoforms significantly increased the multiplicity of liver metastases, whereas CD44v6 and NUMB1/3 did not differ from the parental controls (*Figure 4D–E*).

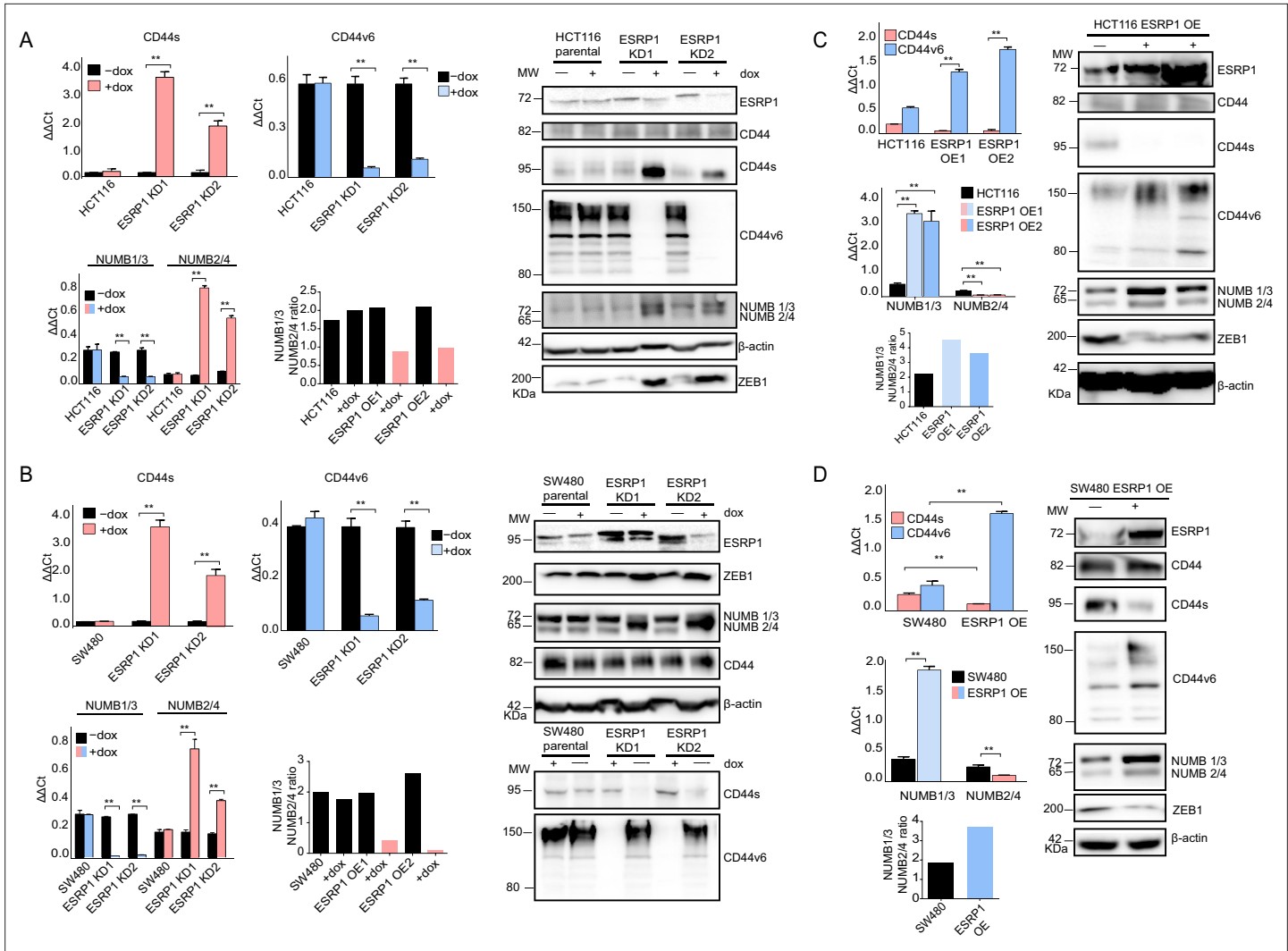

**Figure 3.** *ESRP1* differential expression regulates *CD44* and *NUMB* alternative splicing (AS) isoforms expression. (**A**) RT-qPCR (left histogram panels) and western (right panel) analysis of CD44 and NUMB isoforms expression in *ESRP1*-KD (sh*ESRP1*-transduced) HCT116 cells. Two independent HCT116 *ESRP1*-KD clones were employed. Cells were induced with 1 μg/mL doxycycline for 72 hr before analysis. Expression of the constitutive *CD44* and *NUMB* exons was employed to normalize the results (means ± SEM, n=3). **=p < 0.01. The ratio of NUMB1/3 and NUMB2/4 bands was quantified by ImageJ and shown in bar plot. Please note that the molecular weight of CD44v6 is expected to range between 80 and 150 kDa (*Azevedo et al., 2018*, *Ponta et al., 2003*). β-Actin was used as loading control for western blots. (**B**) RT-qPCR (left histogram panels) and western (right panel) analysis of CD44 and NUMB isoforms expression in *ESRP1*-KD (sh*ESRP1*-transduced) SW480 cells. Two independent SW480 *ESRP1*-KD clones were employed. Cells were induced with 1 μg/mL doxycycline for 72 hr before analysis. Expression of the constitutive *CD44* and *NUMB* exons was employed to normalize the results (means ± SEM, n=3). **=p < 0.01. The ratio of NUMB1/3 and NUMB2/4 bands was quantified by ImageJ and shown in bar plot. Please note that the molecular weight of CD44v6 is expected to range between 80 and 150 kDa (*Azevedo et al., 2018*, *Ponta et al., 2003*). β-Actin was used as loading control for western blots. (**C**) RT-qPCR (left histogram panels) and western (right panel) analysis of CD44 and NUMB isoforms expression in *ESRP1*-OE HCT116 cells. Two independent HCT116 *ESRP1*-OE clones were employed. Expression of the constitutive *CD44* and *NUMB* exons was employed to normalize the results (means ± SEM, n=3). **=p < 0.01. The ratio of NUMB1/3 and NUMB2/4 bands was quantified by ImageJ and shown in bar plot. Please note that the molecular weight of CD44v6 is expected to range between 80 and 150 kDa (*Azevedo et al., 2018*, *Ponta et al., 2003*). β-Actin was used as loading control for western blots. (**D**) RT-qPCR (left histogram panels) and western (right panel) analysis of CD44 and NUMB isoforms expression in *ESRP1*-OE SW480 cells. Expression of the constitutive *CD44* and *NUMB* exons was employed to normalize the results (means ± SEM, n=3). **=p < 0.01. The ratio of NUMB1/3 and NUMB2/4 bands was quantified by ImageJ and shown in bar plot. Please note that the molecular weight of CD44v6 is expected to range between 80 and 150 kDa (*Azevedo et al., 2018*, *Ponta et al., 2003*). β-Actin was used as loading control for western blots.

The online version of this article includes the following source data and figure supplement(s) for figure 3:

**Source data 1.** Original files and labelled bands of western blots in *Figure 3A*.

**Source data 2.** Original files and labelled bands of western blots in *Figure 3B*.

*Figure 3 continued on next page*

*Figure 3 continued*

**Source data 3.** Original files and labelled bands of western blots in *Figure 3C*.

**Source data 4.** Original files and labelled bands of western blots in *Figure 3D*.

**Figure supplement 1.** *ESRP1*, *CD44*, and *NUMB* isoforms analysis in overexpressing and knockdown (KD) colon cancer cell lines.

**Figure supplement 1—source data 1.** Original files and labelled bands of PCR gels in *Figure 3—figure supplement 1A*.

**Figure supplement 1—source data 2.** Original files and labelled bands of PCR gels in *Figure 3—figure supplement 1B*.

**Figure supplement 1—source data 3.** Original files and labelled bands of PCR gels in *Figure 3—figure supplement 1C*.

**Figure supplement 1—source data 4.** Original files and labelled bands of PCR gels in *Figure 3—figure supplement 1*.

**Figure supplement 1—source data 5.** Original files and labelled bands of western blots in *Figure 3—figure supplement 1*.

**Figure supplement 1—source data 6.** Original files and labelled bands of western blots in *Figure 3—figure supplement 1*.

**Figure supplement 2.** *CD44* and *NUMB* isoform-specific expression affects cell migration and Notch signaling activation.

Next, in order to elucidate the signaling pathways and molecular and cellular mechanisms triggered by the CD44 isoforms, we analyzed by RNAseq HCT116 and SW480 cells ectopically expressing CD44s and CD44v6. After dimension reduction with principal component analysis (PCA), the samples separated by group (i.e. CD44s-OE, CD44v6-OE, and controls) (*Figure 5A*). Notably, the CD44s-OE samples showed most distinct expression in both cell lines when compared to the parental and CD44v6-OE cell lines. In HCT116, the CD44v6 samples shared most similarity with the CD44s samples, while in SW480, the CD44v6 samples were most similar to the parental cell line. Thus, we observed both an isoform-independent effect, presumably as the result of the ectopic CD44 expression (and most dominantly visible in HCT116), and an isoform-dependent effect as depicted by the separation of CD44s and CD44v6 samples (*Figure 5A*). As expected, differential expression analysis of the CD44s and v6 isoforms overexpressing samples compared with the parental cell lines revealed an overall upregulation of gene expression (*Figure 5—figure supplement 1A*). Next, in order to identify which genes are specifically upregulated by the different CD44 isoforms, we performed differential expression analysis between the CD44s samples and the CD44v6 samples. To this aim, we employed k-means clustering on the scaled expression values to separate genes specific for the CD44s isoform (e.g. *SPARC*, *ZEB1*, *VIM*), the CD44v6 isoform (e.g. *IL32*, *TACSTD2*, *CSF2*), and genes that were indiscriminative for the CD44v6 isoform or the parental cell lines (e.g. *MAL2*, *ESRP1*, *CDH1*) (*Figure 5B*). Finally, to identify the most distinct differences in signaling pathways and GO functional categories, we performed a gene set enrichment analysis (GSEA) by comparing the CD44s- with the CD44v6-OE samples in the individual cell lines. Among the significantly altered pathways (normalized enrichment score [NES] >1, pval <0.05), EMT was the only one upregulated in CD44s vs. CD44v6 in both cell lines (*Figure 5C–D*). Additional pathways and GO categories activated by CD44s appeared to be cell line specific, for example, Wnt β-catenin signaling (HCT116) and oxidative phosphorylation (SW480). Of note, the detailed GSEA evidenced how several inflammatory (TNF/NFκB; IL6/JAK/STAT3; IFα/γ; ILK2/STAT5) and signaling (KRAS, MYC, E2F) pathways were common to both CD44s and v6, presumably as the result of the ectopic CD44 expression, regardless of the isoform (*Figure 5—figure supplement 1B*).

## Increased ZEB1 and decreased ESRP1 expression correlate with the NUMB2/4 and CD44s isoforms and with poor overall survival

In order to assess the clinical relevance of the results obtained with the SW480 and HCT116 cell lines, we analyzed RNAseq data from patient-derived colon cancers available from the public domain and the scientific literature. To this aim, the TCGA Splicing Variants Database (TSVdb; http://www.tsvdb.com/) was employed to integrate clinical follow-up data with RBP and AS expression profiles obtained from The Cancer Genome Atlas (TCGA) project and from the *Guinney et al., 2015*, study on the classification of human colon cancers into four consensus molecular subtypes (CMS1-4). The main limitation of this approach is the low representation of quasi-mesenchymal (EpCAM$^{lo}$-like) subpopulations in bulk RNAseq preparations and the masking effect that the majority of epithelial (EpCAM$^{hi}$-like) cancer cells are likely to cause. To identify tumors enriched in EpCAM$^{lo}$-like cells, we first stratified them based on *ZEB1* expression (*ZEB1* >8.6: ZEB1$^{hi}$; ZEB1 <8.3. ZEB1$^{lo}$; 8.2<ZEB1<8.6: Intermediate). Subsequently, we used *ESPR1* expression levels to further define the tumors into *ZEB1$^{hi}$ESRP1$^{lo}$*

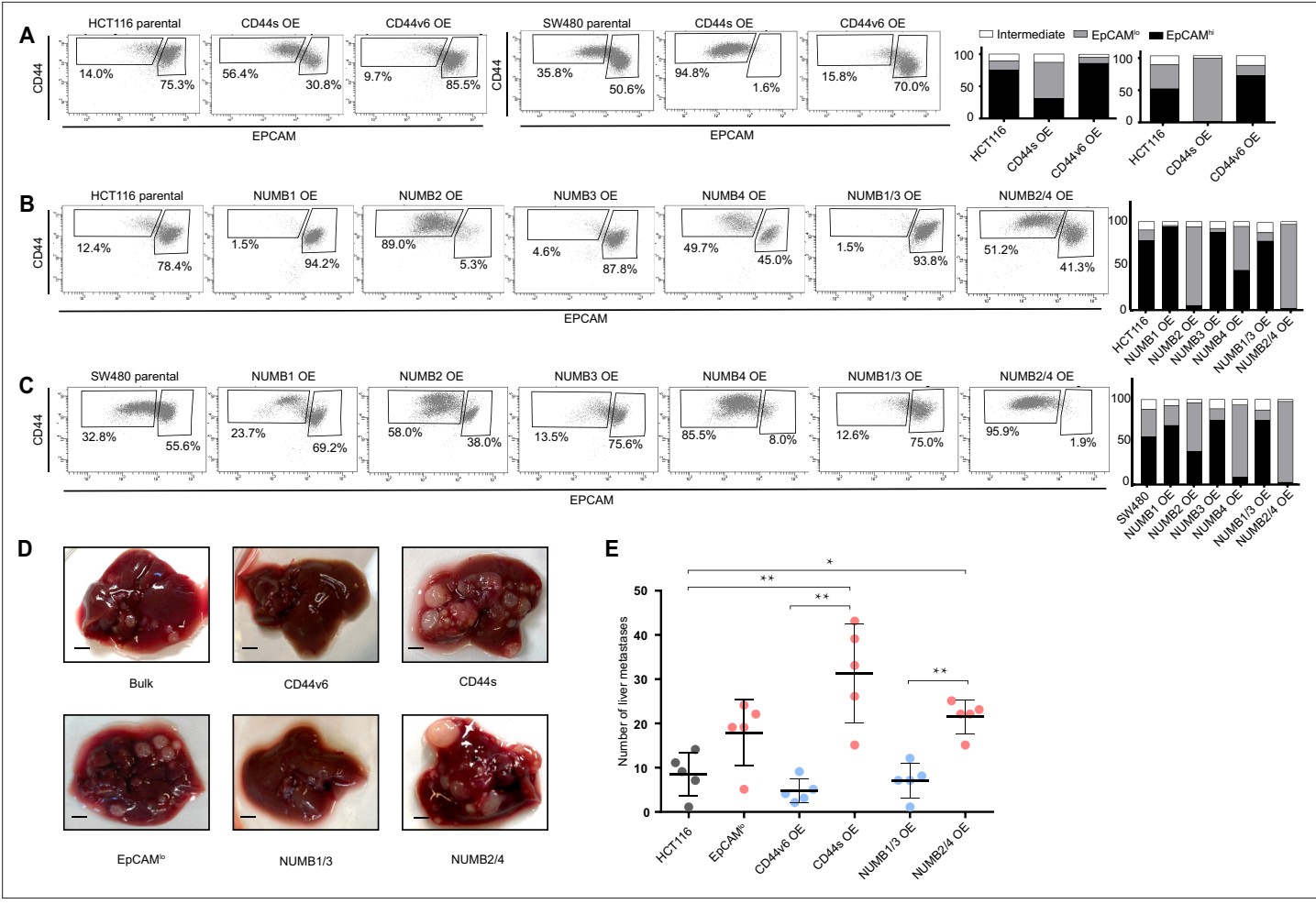

**Figure 4.** *CD44* and *NUMB* alternative splicing (AS) isoforms have opposite functions in quasi-mesenchymal and epithelial colon cancer cells and their capacity to metastasize the liver. (**A**) CD44/EpCAM FACS analysis of EpCAM$^{lo}$ and EpCAM$^{hi}$ subpopulations in CD44s-OE (left) and CD44v6-OE HCT116 and SW480 cell lines. The bar charts on the right depict the percentages of EpCAM$^{lo}$ and EpCAM$^{hi}$ cells. The subpopulation of cells mapping in between, but yet outside, the CD44$^{hi}$EpCAM$^{hi}$ and CD44$^{hi}$EpCAM$^{lo}$ gates, is here labelled as 'intermediate'. (**B**) and (**C**) CD44/EpCAM FACS analysis of EpCAM$^{lo}$ and EpCAM$^{hi}$ subpopulations in NUMB1-4 OE HCT116 and SW480 cells. The bar charts on the right depict the percentages of EpCAM$^{lo}$ and EpCAM$^{hi}$ cells. (**D**) Macroscopic images of livers from mice spleen-injected with CD44s-, CD44v6-, NUMB2/4-, and NUMB1/3-OE HCT116 cells. HCT116 EpCAM$^{lo}$ and bulk cells were used as positive control. Scale bar: 5 mm. (**E**) Liver metastasis multiplicity after intrasplenic injection of CD44s-, CD44v6-, NUMB2/4-, and NUMB1/3-OE HCT116 cells. For each transplantation experiment, 5×10$^4$ cells were injected in the spleen of recipient NSG mouse. Six weeks after injection, mice were sacrificed and individual tumors counted. (means ± SEM, n=5) *=p < 0.05; **=p < 0.01.

The online version of this article includes the following figure supplement(s) for figure 4:

**Figure supplement 1.** CD44 and NUMB isoforms regulate colon cancer cell proliferation.

(*ESRP1* <11.8; hereafter referred to as *ZEB1*$^{hi}$), *ZEB1*$^{lo}$*ESRP1*$^{hi}$ (*ESRP1* >11.6; hereafter referred to as *ZEB1*$^{lo}$). Tumors with intermediate *ZEB1* expression levels and tumors with *ESRP1* expression levels outside these thresholds were defined as intermediate (***Figure 6A***). Kaplan-Meier analysis showed that *ZEB1*$^{hi}$ tumors have an overall decreased survival probability (p=0.045) (***Figure 6B***). Next, we compared the expression of CD44 and NUMB isoforms across the *ZEB1*$^{hi/lo}$ tumors. Notably, while no significant differences were observed based on the expression level of the whole CD44 and NUMB genes, significant differences were found for their specific isoforms (***Figure 6C***). Analysis of the specific isoforms expression across the different CMS (***Guinney et al., 2015***) revealed elevated CD44s and NUMB2/4 expression in the CMS4 subtype, known to be enriched in mesenchymal lineages in tumor and TME cells, and strongly associated with poor survival and the greatest propensity to form distant metastases (***Figure 6D***). Likewise, the majority of the *ZEB1*$^{hi}$ group was composed of the CMS4

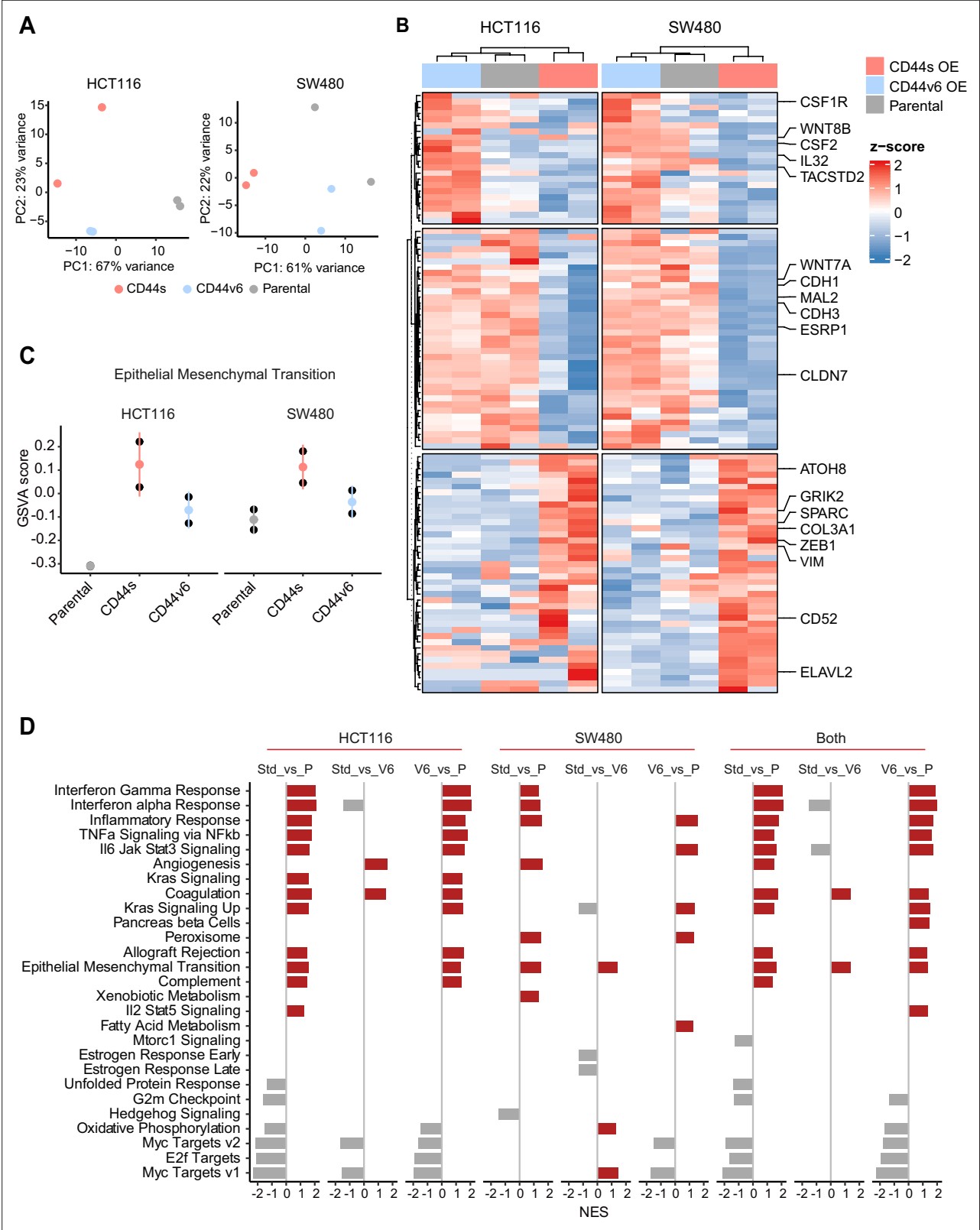

**Figure 5.** RNAseq analysis of CD44s- and CD44v6-expressing colon cancer cells reveals a broad spectrum of downstream alternative splicing (AS) targets and biological functions. (**A**) Principal component analysis (PCA) of RNAseq profiles from CD44s- and CD44v6-OE HCT116 and SW480 cell lines. (**B**) Heatmap of differentially expressed gene among HCT116 and SW480 CD44s-OE, CD44v6-OE, and parental cells. (**C**) Gene set enrichment analysis (GSEA) of epithelial-mesenchymal transition (EMT) in expression profiles from HCT116 and SW480 parental, CD44s-OE, and CD44v6-OE cells.

*Figure 5 continued on next page*

Figure 5 continued

Normalized enrichment score (NES) >1, and pval <0.05. (**D**) GSEA of HCT116 and SW480 expression profiles in parental, CD44s-OE, CD44v6-OE cells compared with each other. Plots show only significantly altered pathways, with NES >1, and pval <0.05.

The online version of this article includes the following figure supplement(s) for figure 5:

**Figure supplement 1.** Gene enrichment and pathway analysis of CD44s- and CD44v6-overexpressing (OE) colon cancer cells.

subtype (72%), while the *ZEB1*[lo] group was mainly contributed by CMS2 (49%) and CMS3 tumors (31%), with few CMS4 tumors (1%) (*Figure 6E*).

Next, we correlated the expression of CD44s/v6 isoforms in patient-derived colon tumors with the DEGs identified in the isoform-overexpressing cell lines (*Figure 7A*). While overall *CD44* expression correlated with both isoforms, the DEGs from the CD44s-OE samples showed specific correlation with CD44s expression in patient-derived tumors (e.g. *SPARC, ZEB1*), the DEGs from the CD44v6 samples correlated with CD44v6 but not with CD44s (e.g. *KDF1, ESRP1*).

Last, we correlated the CD44 and NUMB isoforms expression in patient-derived colon cancers with functional signatures obtained by averaging the scaled expression levels for each of the hallmark sets (*Liberzon et al., 2015*). The CD44s and NUMB2/4 isoforms showed overall similar correlating hallmarks and pathways. However, the same was not true when compared to the CD44v6- and NUMB1/3-associated functional signatures. Here, most invasion/metastasis-relevant hallmarks (e.g.

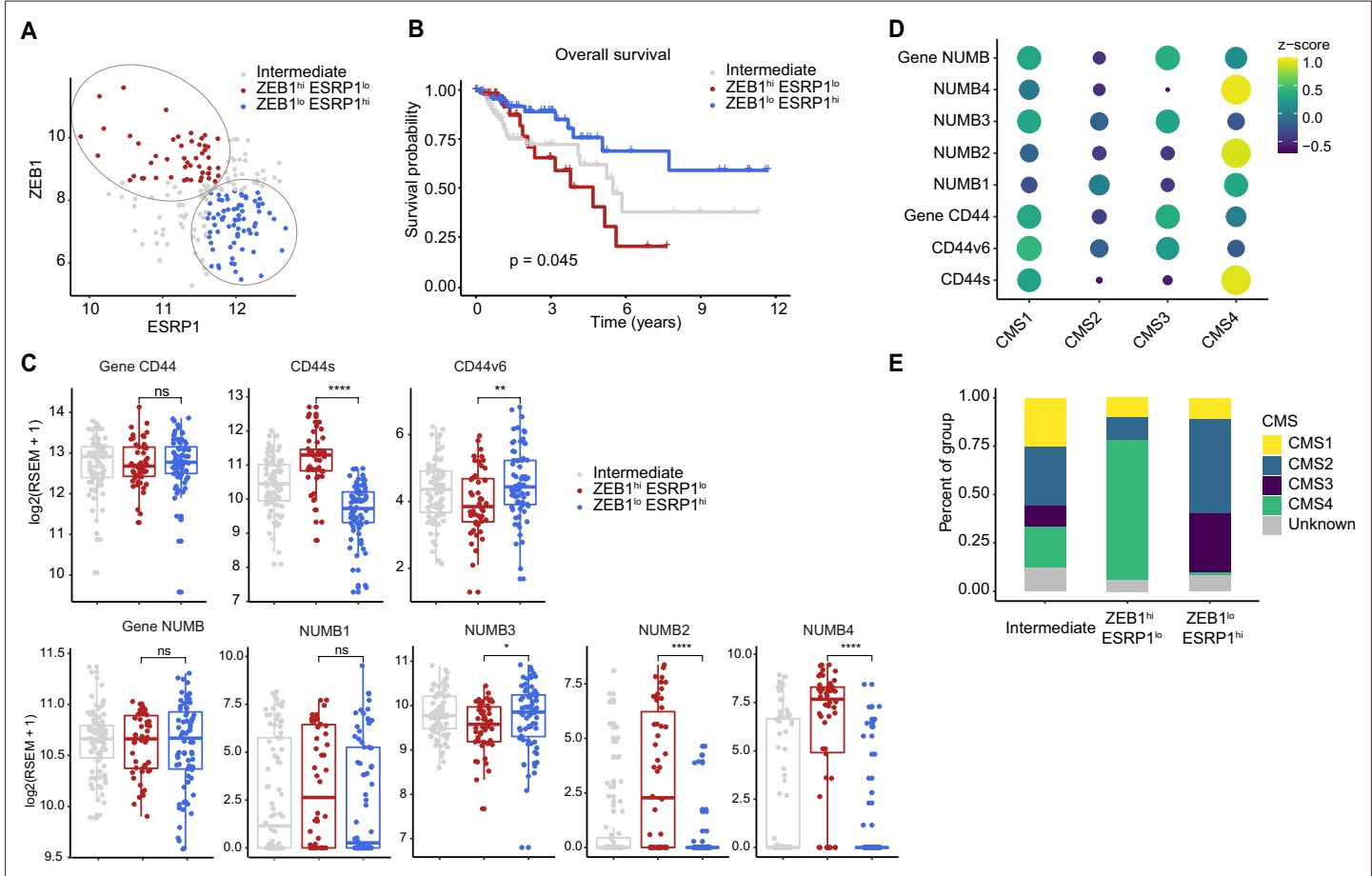

**Figure 6.** Increased *ZEB1* and decreased *ESRP1* expression correlate with the NUMB2/4 and CD44s isoforms and with poor overall survival. (**A**) RNAseq data from The Cancer Genome Atlas (TCGA) were subdivided into three groups based on *ZEB1* and *ESRP1* expression level: *ZEB1*[hi]*ESRP1*[lo] (*ZEB1*[hi], red dots), *ZEB1*[lo]*ESRP1*[hi] (*ZEB1*[lo], blue dots), and intermediate (gray dots). (**B**) Kaplan-Meier analysis of overall survival in the *ZEB1*[hi]*ESRP1*[hi] and *ZEB1*[lo]*ESRP1*[lo] patient groups. (**C**) Box plots showing CD44 and NUMB gene and isoforms expression across the *ZEB1*[hi]*ESRP1*[lo], *ZEB1*[lo]*ESRP1*[hi], and intermediate patient groups. (**D**) Dot plot analysis of the z-score scaled expression values of CD44s, CD44v6, NUMB1-4 isoforms across the four colon cancer consensus molecular subtypes (CMS). (**E**) Stacked bar plot showing the composition of the CMS across the *ZEB1*[hi/lo] and intermediate patient groups.

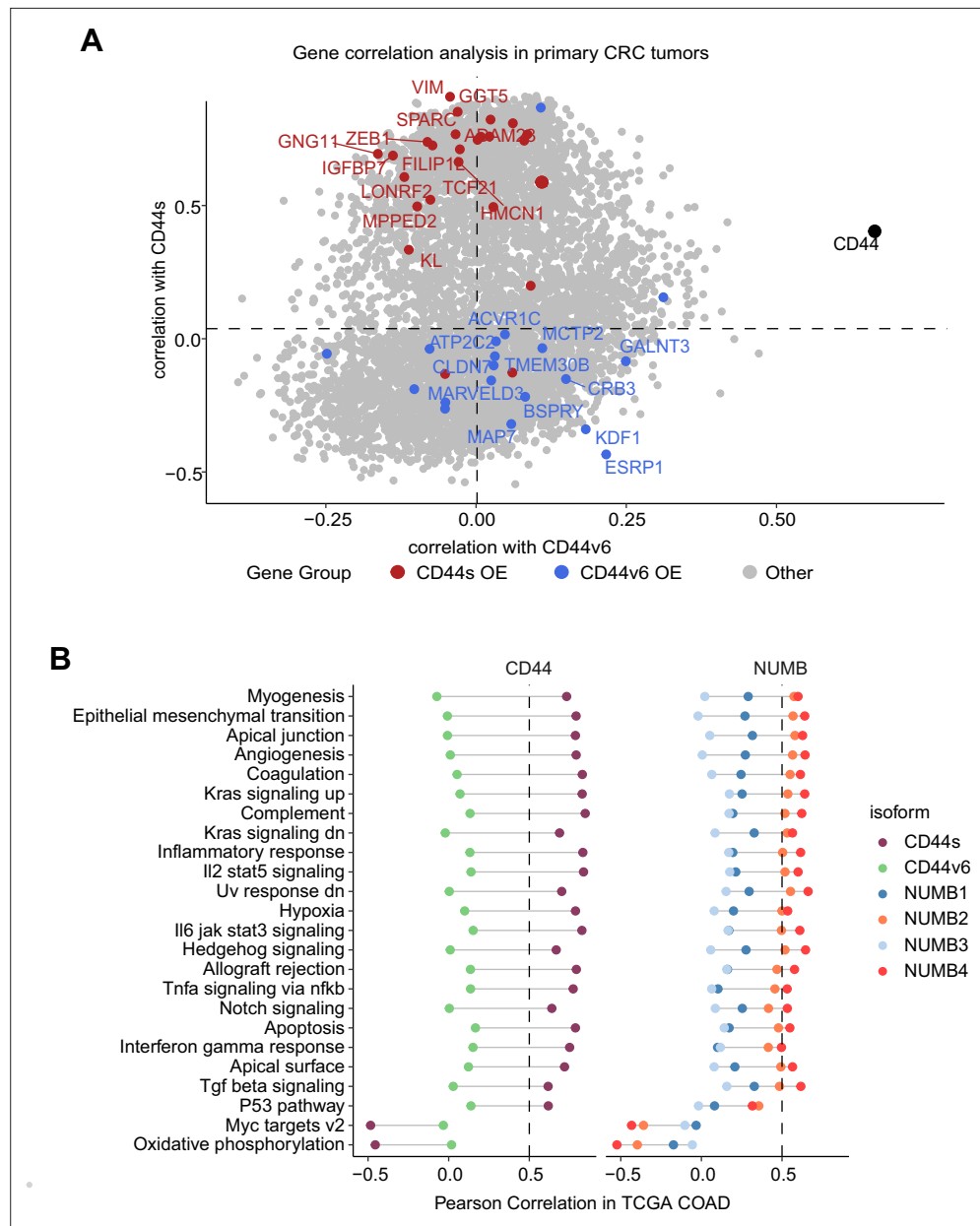

**Figure 7.** Gene and pathway correlation analyses of CD44 and NUMB isoforms in patient-derived colon cancers. (**A**) Gene correlation analysis showing the correlation of gene expression with CD44s and CD44v6 isoform expression in the The Cancer Genome Atlas (TCGA) patient cohort. Differentially expressed genes from CD44s- (red) and CD44v6-OE (blue) RNAseq data are highlighted. (**B**) Pathway correlation analysis showing the correlation of pathway activity CD44 and NUMB isoform expression in the TCGA patient cohort.

The online version of this article includes the following figure supplement(s) for figure 7:

**Figure supplement 1.** *CD44* and *NUMB* isoforms expression in EpCAM$^{hi/lo}$ ovarian and cervical cancer cells.

EMT, angiogenesis, apical junctions) showed a positive correlation with CD44s and NUMB2/4, though not with CD44v6 and NUMB1/3 (**Figure 7B**).

In sum, we confirmed a switch in isoform expression (CD44v6 vs. CD44s and NUMB1/3 vs. NUMB2/4) as a function of *ESRP1* and *ZEB1* expression in colon cancer. Expression of the EpCAM$^{lo}$-specific isoforms (CD44s and NUMB2/4) is elevated in CMS4 tumors overall survival.

## Upregulation of the NUMB2/4 and CD44s isoforms is common to quasi-mesenchymal cells from cancers other than colon

In order to assess whether the preferential expression of the NUMB2/4 and CD44s isoforms is specific to the modalities of local invasion and distant metastasis characteristic of colon cancer, we interrogated expression profiling data previously obtained by comparing epithelial and quasi-mesenchymal subpopulations from ovarian (OV90) and cervical (SKOV6) cancer cell lines (*manuscript in preparation*). Ovarian cancer, because of the distinct anatomical localization of the primary lesion, metastasizes the abdominal cavity with very different modalities than colon cancer, namely by peritoneal dissemination rather than local dissemination into the stroma microenvironment followed by intra- and extravasation of the portal blood stream (*Adam and Adam, 2004*; *Goswami et al., 2009*). On the other hand, metastasis in carcinoma of the cervix occurs both by lymphatic and by hematogenous spread to the lung, liver, and bones. We asked whether, notwithstanding the distinctive patterns of metastatic spread, the CD44s and NUMB2/4 isoforms were preferentially expressed in the corresponding EpCAM$^{lo}$ RNAseq profiles. To this aim, EpCAM$^{hi/lo}$ subpopulations from OV90 and SKOV6 were sorted and analyzed by RNAseq and RT-qPCR, similar to our previous study on colon cancer (*Sacchetti et al., 2021*). As shown in *Figure 7—figure supplement 1*, both NUMB2/4 and CD44s isoforms appear to be upregulated in the OV90 and SKOV6 cell lines, as also validated by RT-qPCR.

## Discussion

The capacity to invade the tumor microenvironment and to form distant metastases undoubtedly represents the most clinically relevant hallmark of epithelial cancer cells. However, the complexity and diversity of the obstacles that carcinoma cells encounter along the invasion-metastasis cascade require transient and reversible changes that cannot be explained by the de novo acquisition of genetic alterations. Instead, epigenetic (non-mutational) modifications underlie phenotypic plasticity, that is, the capacity of cancer cells with a given genotype to acquire more than one phenotype in a context-dependent fashion (*Varga and Greten, 2017*). EMT and MET are central to the phenotypic plasticity characteristic of metastasizing carcinoma cells and are prompted by a broad spectrum of epigenetic mechanisms ranging from chromatin remodeling by histone modifications, DNA promoter methylation, non-coding RNAs, and AS (*Dixit et al., 2016*). Here, we have taken advantage of our previous identification of phenotypic plastic and highly metastatic EpCAM$^{lo}$ colon cancer cells (*Sacchetti et al., 2021*) to characterize the genome-wide AS events that accompany EMT/MET state transitions between the epithelial bulk (EpCAM$^{hi}$) and the quasi-mesenchymal subpopulation.

In view of the central role played by RBPs in eliciting AS, we first identified RBP-coding genes differentially expressed between the EpCAM$^{lo}$ and EpCAM$^{hi}$ fractions of two commonly employed colon cancer cell lines, representative of the chromosomal- and microsatellite-instable subtypes (SW480, CIN; HCT116, MIN) (*Lengauer et al., 1997*). The *ESRP1/2* genes (*Warzecha et al., 2009*), the '*splicing masterminds*' of EMT (*Tavanez and Valcárcel, 2010*; *Warzecha et al., 2010*), were found among the top downregulated RBP-coding genes in EpCAM$^{lo}$ colon cancer cells, as part of a self-enforcing feedback loop with the EMT-TF *ZEB1* (*Preca et al., 2015*). Accordingly, *ZEB1* upregulation in EpCAM$^{lo}$ colon cancer cells is invariably accompanied by *ESRP1/2* downregulation, and *ZEB1*$^{hi}$/*ESRP1*$^{lo}$ colon cancers, predominantly belonging to the mesenchymal CMS4 subgroup, have a significantly worse survival outcome when compared with *ZEB1*$^{lo}$/*ESRP1*$^{hi}$ patients.

Apart from *ESRP1*, several other RBP-coding genes were found to be differentially expressed between epithelial and quasi-mesenchymal colon cancer cells. Whereas the majority of RBP-coding DEGs, like *ESRP1*, appear to be downregulated upon EMT induction (*ESRP1/2*, *RBM14/19/47*, *MBNL3*, *HNRPAB/PF*, *USAF2*), others were activated in the quasi-mesenchymal EpCAM$^{lo}$ fraction (*NOVA2*, *MBNL2*, *QKI*, *SRSF5*, *HNRNPH*, *RBM24/43*). Accordingly, in patient-derived colon cancers stratified according to their consensus molecular signature, the same *QKI*, *RBM24*, and *MBNL2* genes were found to have increased expression in CMS4 tumors, known for their pronounced mesenchymal composition and poor prognosis (*Guinney et al., 2015*). Of note, the mesenchymal nature of CMS4 tumors has previously been questioned as these lesions often feature pronounced infiltration from the surrounding microenvironment, the extent of which might cover their true cellular identity other than representing a mere contamination from the tumor microenvironment (*Calon et al., 2015*; *Isella et al., 2015*). As shown in our previous study (*Sacchetti et al., 2021*), the EpCAM$^{lo}$ cells do represent

bona fide quasi-mesenchymal colon cancer cells, enriched among CMS4 cases, and likely responsible for their poor prognosis. The observed upregulation of RBPs such as quaking (*QKI*) is caused by the presence in its 3'UTR of target sequences of the miR-200 family of microRNAs (*Pillman et al., 2018*; *Kim et al., 2019*). The latter is analogous to the regulation of the expression of the EMT-TF *ZEB1* gene, whose activation during EMT is regulated by the same microRNA family (*Brabletz and Brabletz, 2010*). Accordingly, the significantly reduced levels of all five miR-200 members in EpCAM$^{lo}$ cells (*Sacchetti et al., 2021*) underly the coordinated upregulation of both *ZEB1* and *QKI*.

The here observed *RBM47* downregulation in CMS4 colon cancers is in agreement with a previous report on its decreased protein expression during EMT in association with metastasis in a cohort of primary CRCs (*Rokavec et al., 2017*). On the other hand, the increased expression of other RBP-coding genes such as *RBM24* and *MBNL2* (muscleblind-like 2) in CMS4 tumors and in EpCAM$^{lo}$ cells is in sharp contradiction with their alleged tumor suppressing roles in colon and other cancers (*Xia et al., 2021*; *Lin et al., 2021*). Of note, MBNL2 regulates cancer migration and invasion through PI3K/AKT-mediated EMT (*Lin et al., 2021*) and its overexpression in breast and cancer cell lines inhibits their metastatic potential (*Zhang et al., 2019b*). In contrast to *MBNL2*, *MBNL3*, a distinct member of the muscleblind family, is downregulated in EpCAM$^{lo}$ colon cancer cells, similar to what reported in prostate cancer by *Lu et al., 2015*. *NOVA2*, a member of the Nova family of neuron-specific RBPs, was also upregulated in the quasi-mesenchymal cells from both cell lines, possibly as the result of the differential expression of miR-7-5p (*Xiao, 2019*), as previously shown in non-small cell lung (*Xiao, 2019*) and prostate (*Lu et al., 2015*) cancer. The identification the AS targets downstream of specific RBPs in quasi-mesenchymal cancer cells from different malignancies will likely clarify these apparent contradictions and shed light on the functional roles of distinct members of the splicing machinery in EMT and metastasis.

The spectrum of AS target genes downstream of the RBPs differentially expressed in EpCAM$^{lo}$ colon cancer cells appears extremely broad when it comes to specific cellular processes or signaling pathways. Nonetheless, comparison of our RNAseq data with KD studies of specific RBPs from the public domain (*ESRP1/2 Nieto et al., 2016*, *RBM47 Yang et al., 2016*, and *QKI*; GEO Accession: GSM4677985) allowed us to identify common and unique AS target genes associated with specific downstream effectors. By following this admittedly imperfect approach, the top four AS targets common to all of the above-mentioned RBPs notwithstanding their up- or downregulation in EpCAM$^{lo}$ colon cancer cells, that is, *CTNND1* (δ- or p120-catenin), *LSR* (lipolysis stimulated lipoprotein receptor), *SLK* (STE20-like kinase), and *TCF7L2* (transcription factor 7-like 2, or TCF4) are known regulators and effectors of EMT (*Hernández-Martínez et al., 2019*; *Shimada et al., 2021*; *Conway et al., 2017*; *Karve et al., 2020*), thus pointing to the central role played by AS in the regulation of EMT in the malignant evolution of colon cancer.

Here, we have focused on CD44 and NUMB as two ESRP1-specific AS target genes with well-established functional roles in EMT and in cancer invasion and metastasis. The CD44s and NUMB2/4 isoforms appear to be specifically expressed in quasi-mesenchymal colon cancer cells both from the immortalized cell lines and from patient-derived tumors, with a striking enrichment in the CMS4 subgroup of colon cancer patients. In contrast, the CD44v6 and NUMB1/3 isoforms are preferentially expressed in the epithelial bulk of the tumor. The latter, as far as CD44v6 is concerned, contrasts what previously reported by *Todaro et al., 2014*, where this specific isoform was found to earmark the colon cancer stem cells (CSCs) which underlie metastasis. CD44v6 and other 'variable' CD44 isoforms (CD44v4-10) earmark *Lgr5*$^+$ intestinal stem cells (ISCs), that is, the cells of origin of intestinal tumors, and accordingly promote adenoma formation in vivo (*Zeilstra et al., 2008*; *Zeilstra et al., 2014*; *Misra et al., 2009*). A plausible explanation for the discordant results lies in the epithelial nature of the models employed in the above study and in the requirement of both EMT and MET for the completion of the invasion-metastasis cascade (*Brabletz et al., 2005*). By employing tumor spheres and freshly sorted CD133$^+$ tumor cells, Todaro et al. focused on epithelial CSCs where, as observed in normal ISCs, the CD44v6 isoform is predominantly expressed, and is necessary for EMT to occur upon interaction with c-MET (*Todaro et al., 2014*). The CD44v6 isoform is required for c-MET activation by hepatocyte growth factor (HGF, or scatter factor) (*Orian-Rousseau et al., 2002*) and as such plays an essential role in triggering EMT at the invasive front where tumor cells are exposed to these TME-secreted factors. Our own immunoprecipitation studies confirmed that CD44v6 but not CD44s binds to cMET in response to HGF stimulation (*data not shown*). Therefore, HGF/SF stimulation of

colon cancer cells along the invasive front will trigger the acquisition of quasi-mesenchymal characteristics and the AS-driven switch from CD44v6 to CD44s, the latter unable to bind HGF and as such controlling the extension of EMT activation. The reverse switch will take place upon the activation of the MET necessary for the colonization of the distal metastatic site. From this perspective, both CD44 isoforms are essential for the completion of the invasion-metastasis cascade.

The functional relevance of the CD44s isoforms has been highlighted in malignancies other than colon cancer, namely in prostate (*Lu et al., 2015*) and breast cancer where it activates, among others, PDGFRβ/Stat3 and Akt signaling to promote EMT and CSC traits (*Brown et al., 2011*; *Zhang et al., 2019a*). GO analysis of the RNAseq profiles from colon cancer cells ectopically expressing CD44s highlighted a broader spectrum of signaling pathways likely to underlie EMT. Accordingly, analysis of RNAseq data from primary colon cancers stratified for their CD44s expression revealed an equally broad spectrum of downstream EMT-related biological processes. Of note, among the DEGs identified upon CD44s ectopic expression which correlate with $ZEB1^{hi}/ESRP1^{lo}$ (and CMS4) colon cancers, the *SPARC* gene, a pEMT marker in the $EpCAM^{hi/lo}$ state transitions (*Sacchetti et al., 2021*), was found.

Expression of NUMB2/4 isoforms both in cells lines and in patient-derived colon tumors is associated with signaling pathways and GO categories largely overlapping with those linked to CD44s (and CD44v6 with NUMB1/3), possibly suggesting synergism between AS at these genes. Accordingly, NUMB is involved in a broad spectrum of cellular phenotypes in homeostasis and in cancer where it mainly function as a tumor suppressor (*Pece et al., 2011*). NUMB inhibits EMT by suppressing the Notch signaling pathway. As such, downregulation of NUMB can induce an EMT phenotype in isoform-specific fashion. Analysis of colon cancer cells individually overexpressing each of the four isoforms revealed an increased basal Notch signaling in NUMB2 and -4, as shown by the expression of the 'universal' targets *HES1* and *HEY1*. Instead, ectopic expression of NUMB1/3 resulted in increased transcriptional levels of the more atypical Notch signaling target *ID2*. Although the functional consequences of the NUMB2/4 (and 1/3) isoforms on Notch regulation of EMT are yet unclear, it seems plausible that the complex network of AS targets activated downstream of the RBP-coding DEGs, including CD44, NUMB, and many others as shown here, will eventually lead to the 'just-right' level of plasticity needed to allow both the 'mesenchymalization' during local invasion and systemic dissemination, and the reacquisition of epithelial features at the distant site of metastasis.

Overall, it appears that AS substantially contributes to the epigenetic mechanisms that underlie EMT/MET in cancer metastasis. From this perspective, several aspects of our study are novel: first, the identification of colon cancer-specific AS target genes paralleled by the corresponding RBPs which, when stratified according to the CMS classification of colon cancers, reveal notable differences and consequences on patients' survival. Moreover, the results of the functional analysis of AS at the CD44 gene contrast what previously reported (*Todaro et al., 2014*) and shed new light on the relevance of the standard and v6 isoforms in the migrating CSC model (*Brabletz et al., 2005*). Comparison of the RBP/AS analysis among colon, cervical, and ovarian cancer highlights how, although the majority of AS targets are common to different types of malignancies in RBP-specific fashion, notable differences also exist possibly in reflection of the specific modalities of local dissemination and distal metastasis formation in different cancers. Also, the use of immortalized cell lines for the analysis of epithelial and quasi-mesenchymal tumor cell subpopulations represents an original approach yet based on an 'old-fashioned' laboratory reagent (*Sacchetti et al., 2021*). Finally, the systematic elucidation of the RBPs and AS targets which underlie phenotypic plasticity in different types of cancer will provide novel tumor-specific targets for therapeutic intervention based on small molecule inhibitors and even RNA vaccination.

## Materials and methods

### Key resources table

| Reagent type (species) or resource | Designation | Source or reference | Identifiers | Additional information |
|---|---|---|---|---|
| Cell line (*Homo sapiens*) | HCT116 (adult colorectal carcinoma) | ECACC | Cat# 91091005, RRID:CVCL_0291 | |

*Continued on next page*

*Continued*

| Reagent type (species) or resource | Designation | Source or reference | Identifiers | Additional information |
|---|---|---|---|---|
| Cell line (*Homo sapiens*) | SW480 (adult colorectal carcinoma) | ECACC | Cat# 87092801, RRID:CVCL_0546 | |
| Transfected construct (*Homo sapiens*) | Human-ESRP1 shRNA | Horizon | Cat# V3THS_400802 | Lentiviral construct to transfect express the shRNA |
| Antibody | Anti-human ZEB1 (rabbit monoclonal) | Cell Signaling | Cat# 3396, RRID:AB_1904164 | WB (1.1000) |
| Antibody | Anti-human ESRP1 (rabbit polyclonal) | Thermo Fisher | Cat# PA5-11520, RRID:AB__2899836 | WB (1.1000) |
| Antibody | Anti-human CD44s (mouse monoclonal) | Thermo Fisher | Cat# MA5-13890, RRID:AB_10986810 | WB (1.100) |
| Antibody | Anti-human CD44v6 (mouse monoclonal) | Abcam | Cat# ab78960, RRID:AB_1603730 | WB (1.1000) |
| Antibody | Anti-human NUMB (rabbit monoclonal) | Cell Signaling | Cat# 2756, RRID:AB_2534177 | WB (1.1000) |
| Antibody | Anti-human B-actin (rabbit monoclonal) | Cell Signaling | Cat# 8457,R RID:AB_10950489 | WB (1.2000) |
| Antibody | Anti-mouse CD44-APC (rat monoclonal) | BD Pharmingen | Cat# 559250, RRID:AB_398661 | FACS (1 μg/$10^6$ cells) |
| Antibody | Anti-human EpCAM-FITC (mouse monoclonal) | GeneTex | Cat# GTX30708, RRID:AB_1240769 | FACS (1 μg/$10^6$ cells) |
| Recombinant DNA reagent | ESRP1 cDNA ORF Clone (human) | Sino Biological | Cat# HG13708-UT | |
| Recombinant DNA reagent | pcDNA empty vector (plasmid) | Gift from Ron Smits | | |
| Recombinant DNA reagent | pcDNA-human-CD44s (plasmid) | Gift from Véronique Orian-Rousseau | | |
| Recombinant DNA reagent | pUC57-human-CD44v6 (plasmid) | Gift from Véronique Orian-Rousseau | | |
| Recombinant DNA reagent | pcDNA-human-NUMB1 (plasmid) | Gift from Salvatore Pece | | |
| Recombinant DNA reagent | pcDNA-human-NUMB2 (plasmid) | Gift from Salvatore Pece | | |
| Recombinant DNA reagent | pcDNA-human-NUMB3 (plasmid) | Gift from Salvatore Pece | | |
| Recombinant DNA reagent | pcDNA-human-NUMB4 (plasmid) | Gift from Salvatore Pece | | |
| Recombinant DNA reagent | shZEB1 | *Sacchetti et al., 2021* | Cat# 1864 | |
| Recombinant DNA reagent | pSLIK-Hygro | Addgene | Cat# 25737 | |
| Software, algorithm | R | Seurat, GSVA, MAGIC (*Stuart et al., 2019*; *van Dijk et al., 2018*; *La Manno et al., 2018*) | RRID:SCR_007322, RRID:SCR_021058 | Version 4.0.4 |
| Software, algorithm | Python | Velocyto, scVelo (*La Manno et al., 2018*; *Bergen et al., 2020*) | RRID:SCR_018167, RRID:SCR_018168 | Version 3.8.3 |
| Software, algorithm | STAR | *Dobin et al., 2013* | RRID:SCR_004463 | |
| Software, algorithm | MISO | *Katz et al., 2010*. | RRID:SCR_003124 | |

## Cell cultures

The human colon cancer cell lines HCT116 and SW480, obtained from the European Collection of Authenticated Cell Culture (ECACC), were cultured in DMEM (11965092, Thermo Fisher Scientific) with 10% FBS (Thermo Fisher Scientific), 1% penicillin/streptomycin (Thermo Fisher Scientific, 15140122), and 1% glutamine (Gibco, 25030024), in humidified atmosphere at 37°C with 5% $CO_2$. Both cell lines tested negative for mycoplasma. The identity of each cell line was confirmed by DNA fingerprinting (STR) with microsatellite markers (Amelogenin, CSF1PO, D13S317, D16S539, D5S818, D7S820, THO1, TPOX, vWA, D8S1179, FGA, Penta E, Penta D, D18S51, D3S1358, D21S11) and compared with the analogous data provided by ATCC, EACC, and https://web.expasy.org/cellosaurus/ (data not shown).

## Plasmid transfection and lentiviral transduction

Stable transfection of the *ESRP1* (Sino Biological plasmid # HG13708-UT), *CD44s*, *CD44v6*, and NUMB1-4 (from VOR) expression plasmids was performed using FuGENE HD transfection reagent (Promega, E2311) according to the manufacturer's protocol and selected with Geneticin (Gibco, 10131035). As for the KD constructs, the *ESRP1*-shRNA plasmid (Horizon, V3THS_335722) was packaged by pPAX2 (Addgene # 12260) and pMD2.G (Addgene # 12259) into HEK293T. The virus-containing supernatant was collected 24 hr after transfection, filtered, and employed to infect the HCT116 and SW480 cell line. Selection was applied with 750 ng/mL puromycin (Invivogen, San Diego, CA, USA) or 800 µg/mL of Geneticin selection for 1–2 weeks. The efficiency of overexpression and KD was assessed by qPCR and western blot 48–72 hr after transfection.

## RT-qPCR and PCR analyses

Total RNA was isolated using TRIzol reagent (Thermo Fisher Scientific, 15596018) and was reverse-transcribed using high-capacity cDNA reverse transcription kit (Life Technologies, 4368814), according to the manufacturer's instructions. RT-qPCR was performed using the Fast SYBR Green Master Mix (Thermo Fisher Scientific) on an Applied Biosystems StepOne Plus Real-Time Thermal Cycling Research with three replicates per group. Relative gene expression was determined by normalizing the expression of each target gene to GAPDH. Results were analyzed using the 2-(ΔΔCt) method. To validate isoform switches by RT-PCR, CD44-specific primers were as listed in *Supplementary file 3*.

## Western analysis

Cells were lysed in 2× Laemmli buffer containing 4% sodium dodecyl sulfate (SDS), 48% Tris 0.5 M pH 6.8, 20% glycerol, 18% $H_2O$, bromophenol blue and 10% 1 M DTT, and subjected to SDS-polyacrylamide gel electrophoresis (PAGE), followed by transfer onto polyvinylidene fluoride membranes (Bio-Rad). After blocking with 5% milk in TBS-Tween, the membranes were incubated with primary antibodies against ZEB1 (1.1000, Cell Signaling, #3396), ESRP1 (1.1000, Invitrogen, PA5-11520), CD44s (1.100, Invitrogen, MA5-13890), CD44v6 (1.1000, Abcam, VFF-7), NUMB (1.1000, Cell Signaling, C29G11), and β-actin (1.2000, Cell Signaling, 8547), followed by polyclonal goat anti-mouse/rabbit immuno-globulins horseradish peroxidase-conjugated secondary antibody (Dako) at appropriate dilutions. The signals were detected with Pierce ECT western blotting subtrade (Thermo) using Amersham AI600 (GE Healthcare, Chicago, IL, USA).

## Flow cytometry analysis and sorting

Single-cell suspensions generated in PBS supplemented with 1% FBS were incubated with anti-EpCAM-FITC (1.20, Genetex, GTX30708), and anti-CD44-APC (1.20, BD Pharmingen, 559250) antibodies for 30 min on ice and analyzed on a FACSAria III Cell Sorter (BD Biosciences). CD44hiEpCAMhiand CD44hi-EpCAMlo HCT116 and SW480 cells were sorted and cultured in humidified atmosphere at 37°C with 5% $CO_2$ for 3–5 days before collecting RNA or protein, as previously described (*Sacchetti et al., 2021*). The subpopulation of cells mapping in between the CD44hiEpCAMhi and CD44hiEpCAMlo gates was labelled as intermediate and was further not employed for analysis.

## MTT assay

For MTT assay, 2×10³ HCT116, SW480 parental, CD44v6, CD44s, and NUMB1-4 OE cells were plated in 96-well plates and incubated at 37°C, 5% $CO_2$. Twenty-four hours later, in the culture medium was supplemented with 100 µL 0.45 mg/mL MTT (3-(4,5-dimethylthiazol-2-yl)-2,5-diphenyltetrazolium

bromide; Sigma-Aldrich) and again incubated for 3 hr. The 96-well plates were then centrifuged at 1000 rpm for 5 min and the culture medium removed. MTT formazan precipitates were solubilized with DMSO. OD reading was performed at 595 nm with microplate reader (Model 550, Bio-Rad). Background measurements were subtracted from each data point. Experiments were performed in duplicate for each individual cell line and drug. Cell numbers were calculated every 24 hr for a 6-day period for proliferation analysis.

## Cell migration assay

Migration assays were conducted with 8 µm pore PET transwell inserts (BD Falcon) and TC-treated multi-well cell culture plate (BD Falcon). $5 \times 10^4$ cells were seeded in 100 µL of serum-free growth medium in the top chamber. Growth medium containing 10% FBS was used as a chemoattractant in the lower chamber. After 24 hr, cells migrated to the lower chamber were fixed with 4% PFA, stained with 0.1% trypan blue solution, and counted under the microscope.

## Mouse spleen transplantation

All mice experiments were implemented according to the Code of Practice for Animal Experiment in Cancer Research from the Netherlands Inspectorate for Health Protections, Commodities and Veterinary Public Health. Mice were fed in the Erasmus MC animal facility (EDC). NOD.Cg-Prkdc$^{scid}$ Il2rg$^{tm1Wjl}$/SzJ (NSG) mice from 8 to 12 weeks of age were used for spleen transplantation. Anesthetics Ketamine (Ketalin, 0.12 mg/mL) and xylazine (Rompun, 0.61 mg/mL) were given intraperitoneally, while the analgesic Carpofen (Rimadyl, 5 mg/mL) was injected subcutaneously. $5 \times 10^4$ HCT116 and SW480 cells resuspended in 50 µL PBS were injected into the exposed spleen with an insulin syringe and left for 15 min before splenectomy. Transplanted mice were sacrificed after 4 and 8 weeks and analyzed for the presence of liver metastases.

## AS analysis

The following public available RNASeq (SRA database) data relative to RBP KD studies were used: ESRP1-KD and RMB47-KD in the human non-small cell lung cancer cell line H358 (*Yang et al., 2016*) with accession ID SRP066789 and SRP066793; ESRP2-KD in the human prostate adenocarcinoma cancer cell line LNCaP (*Nieto et al., 2016*) with accession ID SRP191570; the QKI-KD in the oral squamous cell carcinoma cell line CAL27 datasets with accession number SRX8772405. Together with our own EpCAM$^{hi/lo}$ RNASeq data obtained from the colon cancer cell lines (*Sacchetti et al., 2021*), the sequencing reads were mapped to GRCh37.p13.genome by STAR (*Dobin et al., 2013*) (https://www.gencodegenes.org/human/release_19.html). MISO (*Katz et al., 2010*) was used to quantify AS events with annotation from https://miso.readthedocs.io/en/fastmiso/index.html#iso-centric. The MISO uses the alternative exon reads and adjacent conservative reads to measure the percentage of transcript isoform with specific exon included, termed PSI or $\Psi$. The PSI ranges from 0 (i.e. no isoform includes a specific alternative exon) to 1 (i.e. all of the isoforms detected comprise the alternative exon).

We removed alternative events with low expression of related transcript isoforms if less than three samples in a dataset had more than 10 informative reads to calculate the PSI. Next, we compared the PSI between RBPs KD and wild type in each cell line, as well as the PSI between EpCAM$^{hi}$ and EpCAM$^{lo}$ groups in the SW480 and HCT116 colon cancer cell lines. AS events were defined as differentially spliced events when the difference of mean PSI between two groups ($\Delta$PSI) was >10%.

## RNAseq analysis

RNA quality was first evaluated by NanoDrop and further purified by DNAse treatment followed by the TURBO DNA-free Kit protocol (Invitrogen). Samples were sequenced with the DNA nanoball (DNB) seq protocol (BGI) to a depth of 50 million reads per sample. Adapter sequences and low-quality sequences were filtered from the data using SOAPnuke software (BGI). Reads were aligned to the human reference genome build hg19 with the RNAseq aligner STAR (v2.7.9a) and the *Homo sapiens* GENCODE v35 annotation. Duplicates were marked with Sambamba (0.8.0) and raw counts were summed using FeatureCounts (subread 2.0.3). Downstream analysis was performed in R using the DESeq2 package (v1.30.1). After variance stabilizing transformation, PCA was performed on each cell line separately. DEGs were identified by comparing the different groups of ectopically expressing CD44 samples with a Wald test, and by selecting the genes with absolute log fold change

above 1.5 and padj <0.1. GSEA was performed with the Fsgsea package using the HallMark geneset from the molecular signature database, and by selecting significant pathways based on NES >1 and p-value <0.05.

## RNAseq data from primary (patient-derived) colon cancers

Patient data from TCGA, with annotation of the CMS as described in *Guinney et al., 2015*, were integrated with splicing data from the TSVdb (http://www.tsvdb.com/). For splicing analysis, RNAseq by expectation maximization values were log transformed and expression levels of each isoform (CD44std: isoform_uc001mvx, CD44v6: exon_chr11.35226059.35226187, NUMB1. isoform_uc001xny, NUMB2. isoform_uc001xoa, NUMB3: isoform_uc001xnz, NUMB4: isoform_uc001xob) were annotated to the patients. Isoform expression was compared in groups based on the CMS groups and tumor expression levels (*ZEB1*, *ESRP1*). Tumors were stratified on *ZEB1* expression levels using a log rank test top optimize overall survival differences (thresholds: 8.3, 8.6). Next, ESRP1 expression was used to purify the groups into ZEB1$^{hi}$ESRP1$^{lo}$ and ZEB1$^{lo}$ESRP1$^{hi}$ (thresholds: 11.6, 11.8). Survival analysis was done using the Kaplan-Meier method with the survival and survminer packages in R. Correlation analysis was done by computing the Pearson correlation between the isoforms and whole gene expression levels as processed in *Guinney et al., 2015*. Likewise, association between isoform expression and pathway activity was evaluated by computing the Pearson correlation between the isoforms and the average scaled expression values of the pathways, as defined in the HallMark gene set from the molecular signature database (*Liberzon et al., 2015*).

## Acknowledgements

We are grateful to Dr Juan Valcarcel (CRG, Barcelona, Spain) for his critical reading of the manuscript.

## Additional information

### Funding

| Funder | Grant reference number | Author |
| --- | --- | --- |
| China Scholarship Council | 201806300047 | Tong Xu |

The funders had no role in study design, data collection and interpretation, or the decision to submit the work for publication.

### Author contributions

Tong Xu, Resources, Validation, Investigation, Visualization, Methodology, Writing - original draft; Mathijs Verhagen, Data curation, Software, Formal analysis, Validation, Methodology; Rosalie Joosten, Resources, Supervision, Methodology; Wenjie Sun, Formal analysis, Methodology; Andrea Sacchetti, Investigation, Methodology; Leonel Munoz Sagredo, Investigation, Methodology, Writing – review and editing; Véronique Orian-Rousseau, Conceptualization, Methodology, Writing – review and editing; Riccardo Fodde, Conceptualization, Resources, Formal analysis, Supervision, Funding acquisition, Validation, Investigation, Methodology, Writing - original draft, Project administration, Writing – review and editing

### Author ORCIDs

Tong Xu http://orcid.org/0000-0002-7046-5917
Mathijs Verhagen http://orcid.org/0000-0003-3126-8379
Riccardo Fodde http://orcid.org/0000-0001-9839-4324

### Ethics

All mice experiments were implemented according to the Code of Practice for Animal Experiment in Cancer Research from the Netherlands Inspectorate for Health Protections, Commodities and Veterinary Public Health. Permit number AVD1010020171344.

### Decision letter and Author response

Decision letter https://doi.org/10.7554/eLife.82006.sa1

Author response https://doi.org/10.7554/eLife.82006.sa2

## Additional files

### Supplementary files

• Supplementary file 1. List of alternative splicing targets in ESRP1 knocking down H358 line (a), HCT116 (b), and SW480 (c) EpCAM^lo and EpCAM^hi subpopulation, filtered by ΔPSI (differential percentage spliced in) > 0.1.

• Supplementary file 2. List of alternative splicing targets in *ESRP1*-KD in the H358 cell line, *ESRP2*-KD in LNCaP, *RBM*47-KD in H358 line, QKI-KD in CAL27, and HCT116 and SW480 EpCAM^lo and EpCAM^hi subpopulation, filtered by ΔPSI (differential percentage spliced in) > 0.1.

• Supplementary file 3. Lists of primer sequences used for RT-PCR analysis.

• Supplementary file 4. Differential expressed gene lists from the RNAseq analysis HCT116 CD44s- and CD44v6-OE cells.

• Supplementary file 5. Differential expressed gene lists from the RNAseq analysis SW480 CD44s- and CD44v6-OE cells.

• Supplementary file 6. List of gene set enrichment analysis (GSEA) in CD44s OE vs. CD44v6 OE vs. parental HCT116 and SW480 cells.

• Supplementary file 7. List of gene set variation analysis (GSVA) in CD44s OE vs. CD44v6 OE vs. parental HCT116 and SW480 cells.

• MDAR checklist

### Data availability

The RNA-sequencing data from this study have been submitted to the Gene Expression Omnibus (GEO) database under the accession number GSE192877. Other data referenced in this study are publicly available and can be accessed from the GEO using GSE154927, GSE154730 and Synapse using identifier syn2623706 .

The following dataset was generated:

| Author(s) | Year | Dataset title | Dataset URL | Database and Identifier |
|---|---|---|---|---|
| Xu T, Verhagen MP, Fodde R | 2022 | CD44s and CD44v6 overexpressed RNAseq profiles of colon cancer cell lines HCT116 and SW480 | https://www.ncbi.nlm.nih.gov/geo/query/acc.cgi?acc=GSE192877 | NCBI Gene Expression Omnibus, GSE192877 |

The following previously published datasets were used:

| Author(s) | Year | Dataset title | Dataset URL | Database and Identifier |
|---|---|---|---|---|
| Sacchetti A | 2020 | CD44highEpCAMhigh and CD44highEpCAMlow RNAseq profiles of colon cancer cell lines HCT116 and SW480, in triplicate | https://www.ncbi.nlm.nih.gov/geo/query/acc.cgi?acc=GSE154927 | NCBI Gene Expression Omnibus, GSE154927 |
| Kang K, Hong JH, Ahn Y, Ko YH | 2020 | Transcriptome datasets of the Quaking (QKI) gene knock-down human oral squamous cell carcinoma (OSCC) cells | https://www.ncbi.nlm.nih.gov/geo/query/acc.cgi=GSE154730 | NCBI Gene Expression Omnibus, GSE154730 |
| Guinney J | 2015 | Colorectal Cancer Subtyping Consortium (CRCSC) | https://doi.org/10.7303/syn2623706 | Synapse, 10.7303/syn2623706 |

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
