## [Editor Report]

This fundamental study provides a valuable analysis of the splicing landscape in colon cancer cells that have properties intermediate between those typically found in primary cancers ("epithelial") and those that are spreading by metastasis ("mesenchymal"). The strength of evidence provided is solid and convincing and supports current ideas that changes in the way that RNA from particular genes is processed plays a key role in cancer spread.

---

## [Decision Letter]

**Decision letter after peer review:**

Thank you for submitting your article "Alternative splicing downstream of EMT enhances phenotypic plasticity and malignant behavior in colon cancer" for consideration by *eLife*. Your article has been reviewed by 2 peer reviewers, and the evaluation has been overseen by a Reviewing Editor and Erica Golemis as the Senior Editor. The reviewers have opted to remain anonymous.

Essential revisions:

1) The authors must more clearly articulate how this paper will move the field on from what is already known from previous work in other cancers and/or to articulate how the work specifically advances our knowledge related to colon cancer.

2) Include Western analyses of the over-expressing and knockdown cells to see how much these interventions changed expression of ESRP1 and ZEB1, and the different splice isoforms relative to the endogenous protein expression.

3) The authors should strengthen the impact of the manuscript by emphasising better that by looking at splicing regulation in all these previously understudied cancers (in terms of EMT) with these very informative quasi-mesenchymal cell lines, their data suggest previously reported changes in splicing environments play a broad and general role on EMT across cancer types.

4) Please ensure that the figures are described sequentially

5) There are several discrepancies in the data. For example, in supplementary figure 1B, the FACS data clearly demonstrates that knockdown of ESRP1 affects CD44 expression, contradicting the statement written on page 11 line 15 that ESRP1 only marginally affects CD44 expression. This is very evident in the SW480 cell line data. Moreover, the two hairpins used to downregulate ESRP1 show variable results- ESRP1 KD-1 negatively affects CD44 whereas ESRP1 KD-2 positively affects CD44. Finally as per the western blot analysis of ESRP1 knockdown performed in SW480 cells (figure 3B), CD44 v6 expression is promoted and CD44s is suppressed and in figure 3C and figure 3D, ESRP1 overexpression is leading to upregulation of both NUMB 1/3 and 2/4 isoforms in HCT116 and SW480 cells as per the Western blot analysis. These discrepancies should be described and carefully addressed. It may be that this is due to differences between cell lines. Hence, additional models should be added to determine the extent of variability across cell types.

6) Please justify normalize specific spliced isoforms of CD44 and NUMB to their constitutive exons of respective genes.

*Reviewer #1 (Recommendations for the authors):*

The authors could state more clearly at the very start the hypothesis/hypotheses they were testing, and how testing these will move the field on from what is already known from previous work in other cancers or in just colon cancer. This would help the reader evaluate success of their work and appreciate the knowledge gap they have filled. Hypothesis was not detected using a word search in the current manuscript.

The quality of the science seems high. While the results support the conclusions, it would have been useful to have Western analyses of the over-expressing and knockdown cells to see how much these interventions changed expression of ESRP1 and ZEB1, and the different splice isoforms relative to the endogenous protein expression.

The extensive list of metastasis related RBPs and splicing targets mentioned at the end of the introduction could be more clearly presented as supplementary tables.

While this study mainly tackles the very important topic of colon cancer, it failed to fully articulate how general their conclusions might be. I thought the impact of this report could be increased if the authors could from the start put their study in a broader context. While reading it I had a feeling that the authors were showing similar outcomes in colon cancer to those already in the literature for cancers like breast. However, the results were actually broader, and generalised to include colon cancer, ovarian cancer or cervical. I wondered whether the authors could strengthen the impact of the manuscript by emphasising better that by looking at splicing regulation in all these previously understudied cancers (in terms of EMT) with these very informative quasi-mesenchymal cell lines, their data suggest previously reported changes in splicing environments play a broad and general role on EMT across cancer types. They do mention this broader context, but the data for other cancers that would increase the scope of these observations was supplementary, and the title itself only mentioned colon cancer.

*Reviewer #2 (Recommendations for the authors):*

After careful evaluation of the manuscript, the authors are urged to address the following questions.

C1: The figures are not cited sequentially in the text. Authors are requested to reorganize their figure panels (both main and supplementary) so that the related experiments are arranged under one common figure panel.

C2: In figure 1D and figure 1E, alterations in ESRP1 expression affects transcription of Zeb1. Similarly, in supplementary figure 4C, CD44s, NUMB2, NUMB4 and NUMB2/4 isoform overexpression is also shown to affect Zeb1. As ESRP1, CD44 and NUMB are downstream targets of Zeb1, why is Zeb1 expression is affected upon modulating the expression or alternative splicing of its effector molecules?

C3: In supplementary figure 1B, the FACS data clearly demonstrates that knockdown of ESRP1 affects CD44 expression, contradicting the statement written on page 11 line 15 that ESRP1 only marginally affects CD44 expression. This is very evident especially from the SW480 cell line data. Moreover, the two hairpins used to downregulate ESRP1 show variable results- ESRP1 KD-1 negatively affects CD44 whereas ESRP1 KD-2 positively affects CD44.

C4: Why only exon skipping type of alternatively spliced events were selected for investigation needs justification. Furthermore, in RT-qPCR analyses, GAPDH is used as control. For understanding the relative abundance of specific spliced isoforms of CD44 and NUMB, their expression should also be normalized with Ct value of the constitutive exons of respective genes.

C5: There is severe inconsistency in the mRNA and protein data in figure 3. As per the western blot analysis of ESRP1 knockdown performed in SW480 cells (figure 3B), CD44 v6 expression is promoted and CD44s is suppressed. Moreover, the knockdown efficiency should also be demonstrated by providing data of ESRP1 bands. In figure 3C and figure 3D, ESRP1 overexpression is leading to upregulation of both NUMB 1/3 and 2/4 isoforms in HCT116 and SW480 cells as per the western blot analysis.

C6: As the contributions of CD44v6 and CD44s in influencing EMT are vastly different, why HCT116 CD44v6 OE cells are more similar to HCT116 CD44s OE cells (than the parental cells which are naturally more epithelial in nature)? Ref: Page 16 line 8.

---

## [Author Response]

Essential revisions:1) The authors must more clearly articulate how this paper will move the field on from what is already known from previous work in other cancers and/or to articulate how the work specifically advances our knowledge related to colon cancer.

We understand and agree on that our paper is not entirely original as similar analyses have been conducted in other cancer types with only apparently similar results. Notwithstanding the latter, our contribution does represent a concrete advancement in our understanding of the relative role played by alternative splicing as one of the main mechanisms underlying EMT/MET in epithelial cancers in general and in colon cancer in particular. There are several reasons for this: first, our study is not limited to the identification of downstream AS target genes but it also extends to the upstream RNA-binding proteins (not only the well-studied ESRP1) which, when stratified according to the CMS classification of colon cancers, reveals notable differences and consequences on patients’ survival. And even when it comes to the individual AS targets chosen for validation and more in-depth functional analysis, our results sharply contrast with what previously reported in the literature on the role of the CD44 standard and v6 isoforms, and of the less characterized NUMB1/3 vs. 2/4 isoforms in EMT. Comparison with our own –admittedly limited- AS analysis of cervical and ovarian cancers, and with additional data from the public domain, highlights how, although the majority of AS targets are common to different types of malignancies in RBP-specific fashion, notable differences also exist possibly in reflection of the specific modalities of local dissemination and distal metastasis formation in different cancers. Last, the use of immortalized cell lines for the sorting of epithelial and quasi-mesenchymal tumor cell subpopulations represents an entirely novel approach yet based on an “old-fashioned” laboratory reagent.

In view of the above, it is not easy to concisely highlight how our manuscript will move the field on from what is already known in other cancers. Nonetheless, we have tried to clarify these issues in the Introduction and Discussion sections of the revised manuscript.

2) Include Western analyses of the over-expressing and knockdown cells to see how much these interventions changed expression of ESRP1 and ZEB1, and the different splice isoforms relative to the endogenous protein expression.

In the revised manuscript, western blots have been added next to the previously included RTqPCRs analyses, to show how *ESRP1* and *ZEB1* ablation and overexpression affect each other at the protein level. Moreover, the same proteins were analyzed by western blots upon CD44 and NUMB isoform knockdown and overexpression (see Figure 1, Figure 3, and Figure 3—figure supplement 1 in the revised manuscript).

3) The authors should strengthen the impact of the manuscript by emphasising better that by looking at splicing regulation in all these previously understudied cancers (in terms of EMT) with these very informative quasi-mesenchymal cell lines, their data suggest previously reported changes in splicing environments play a broad and general role on EMT across cancer types.

See above (point no. 1). We will strengthen these aspects in the Discussion section of the revised manuscript.

4) Please ensure that the figures are described sequentially

We apologize for this. In the revised manuscript the figures are now described sequentially.

5) There are several discrepancies in the data. For example, in supplementary figure 1B, the FACS data clearly demonstrates that knockdown of ESRP1 affects CD44 expression, contradicting the statement written on page 11 line 15 that ESRP1 only marginally affects CD44 expression. This is very evident in the SW480 cell line data. Moreover, the two hairpins used to downregulate ESRP1 show variable results- ESRP1 KD-1 negatively affects CD44 whereas ESRP1 KD-2 positively affects CD44. Finally as per the western blot analysis of ESRP1 knockdown performed in SW480 cells (figure 3B), CD44 v6 expression is promoted and CD44s is suppressed and in figure 3C and figure 3D, ESRP1 overexpression is leading to upregulation of both NUMB 1/3 and 2/4 isoforms in HCT116 and SW480 cells as per the Western blot analysis. These discrepancies should be described and carefully addressed. It may be that this is due to differences between cell lines. Hence, additional models should be added to determine the extent of variability across cell types.

We are grateful to the reviewers for having spotted these discrepancies in the data which are exclusively the result of our own mistakes in the figures’ illustrations.

As for Figure 1F (*ESRP1* OE) compared with Figure 1—figure supplement 1B (*ESRP1* KD), it is evident that the former gives rise to much clearer results than the latter. This said, the statement according to which both OE and KD of *ESRP1* have limited quantitative effects on CD44 expression levels (the height of the cloud of cells along the y-axis remains approx. the same between parental and OE/KD samples) is correct. This is in contrast with EpCAM (the x-axis) where the increase in expression levels is evident in the OE clones. The opposite effect is not as evident in the KD clones where, as also pointed out by the reviewers, variable results are observed between individual clones (see KD1 vs KD2 in SW480). FACS analysis of a large number of individual clones (not shown), does confirm that the KD reagents are not very efficient especially in the SW480 cell line (see Figure 1E). In the revised manuscript, we have now added the FACS analysis of the pooled KD1 and KD2 transfections which do not suffer from the intra clones variability and accordingly depicts a more coherent behavior in both cell lines (i.e. significant EpCAM downregulation; see Figure 1—figure supplement 1B). If preferred, we can also add a supplementary figure showing tens of clones to highlight the variability of the results at the individual clone level. Apart from the variable efficacy of the shRNA reagents in the two cell lines, these data are interpreted as the result of the interference of RBPs other than ESRP1 in the regulation of EMT/MET in the cell lines. Overexpression of ESRP1 instead, has dominant effects and results in more clear differences (mainly in EpCAM) with the parental lines.

In the revised manuscript, the sentence on page 11 of the revised manuscript now reads: “Of note, ESRP1-overexpression in the HCT116 and SW480 cell lines resulted in the dramatic reduction of their EpCAM^lo^ subpopulations and the expansion of the epithelial bulk (EpCAM^hi^), as shown by FACS analysis (Figure 1F, Figure 1—figure supplement 1A). However, ESRP1 knockdown (KD) gave rise to less clear and extremely variable results among the individual clones analyzed by FACS, in particular in the SW480 cell line. More coherent and representative result were obtained with the pools of the KD transfections (Figure 1—figure supplement 1B). These results suggest that RNA binding proteins other than ESRP1 are likely to be involved in the alternative splicing regulation of the EpCAM^lo^ colon cancer subpopulation.”

As for the western analysis of the *ESRP1* KD in SW480 cells, we again apologize for our own mistake while preparing Figure 3B: in the last panel (far right) the plus and minus signs relative to the Dox-driven induction have been inverted thus leading to the apparently opposite effects than expected. This has now been corrected in the revised manuscript. We also added bar plot graphs to Figure 3A-D and Figure 3—figure supplement 1H showing the ratio between NUMB1/3 and NUMB2/4, as analyzed by ImageJ on the western blot bands.

We truly apologize for the confusion we generated and are grateful to the reviewers and editors for having pointed out to us these discrepancies.

6) Please justify normalize specific spliced isoforms of CD44 and NUMB to their constitutive exons of respective genes.

The normalization of the specific isoforms with GAPDH is justified by our previous analysis of its constant expression levels throughout the samples. However, to satisfy the reviewer’s request, we have now modified the manuscript so that all normalizations of the expression values of the various CD44 and NUMB isoforms have been calculated based on their constitutive exons (see Figure 2, Figure 3, and Figure 3—figure supplement 1). As shown, the results do not change when compared with those obtained by GAPDH normalization.

Reviewer #1 (Recommendations for the authors):The authors could state more clearly at the very start the hypothesis/hypotheses they were testing, and how testing these will move the field on from what is already known from previous work in other cancers or in just colon cancer. This would help the reader evaluate success of their work and appreciate the knowledge gap they have filled. Hypothesis was not detected using a word search in the current manuscript.The quality of the science seems high. While the results support the conclusions, it would have been useful to have Western analyses of the over-expressing and knockdown cells to see how much these interventions changed expression of ESRP1 and ZEB1, and the different splice isoforms relative to the endogenous protein expression.The extensive list of metastasis related RBPs and splicing targets mentioned at the end of the introduction could be more clearly presented as supplementary tables.While this study mainly tackles the very important topic of colon cancer, it failed to fully articulate how general their conclusions might be. I thought the impact of this report could be increased if the authors could from the start put their study in a broader context. While reading it I had a feeling that the authors were showing similar outcomes in colon cancer to those already in the literature for cancers like breast. However, the results were actually broader, and generalised to include colon cancer, ovarian cancer or cervical. I wondered whether the authors could strengthen the impact of the manuscript by emphasising better that by looking at splicing regulation in all these previously understudied cancers (in terms of EMT) with these very informative quasi-mesenchymal cell lines, their data suggest previously reported changes in splicing environments play a broad and general role on EMT across cancer types. They do mention this broader context, but the data for other cancers that would increase the scope of these observations was supplementary, and the title itself only mentioned colon cancer.

We already replied relative to these issues in the 1^st^ of the “essential revisions” (see above). As for the lists of metastasis related RBPs and splicing targets mentioned at the end of the introduction, we are not sure what exactly the reviewer is requesting. The lists were already included in the original manuscript as Supplementary Tables in Excel format. Moreover, a more specific list was already included in the original manuscript (Figure 1—figure supplement 1C).

Reviewer #2 (Recommendations for the authors):After careful evaluation of the manuscript, the authors are urged to address the following questions.C1: The figures are not cited sequentially in the text. Authors are requested to reorganize their figure panels (both main and supplementary) so that the related experiments are arranged under one common figure panel.

We apologize for this. In the revised manuscript the figures are now described sequentially.

C2: In figure 1D and figure 1E, alterations in ESRP1 expression affects transcription of Zeb1. Similarly, in supplementary figure 4C, CD44s, NUMB2, NUMB4 and NUMB2/4 isoform overexpression is also shown to affect Zeb1. As ESRP1, CD44 and NUMB are downstream targets of Zeb1, why is Zeb1 expression is affected upon modulating the expression or alternative splicing of its effector molecules?

The relationship between ESRP1 and ZEB1 is not a conventional epistatic one with an upstream signal and a downstream effector. Instead, as pointed out in the Results section, it has been shown that in breast and pancreas cancer *ZEB1­*-driven EMT downregulates the expression *ESRP1* as part of a self-enforcing feedback loop (*Int J Cancer*, 2015, 137:2566).

C3: In supplementary figure 1B, the FACS data clearly demonstrates that knockdown of ESRP1 affects CD44 expression, contradicting the statement written on page 11 line 15 that ESRP1 only marginally affects CD44 expression. This is very evident especially from the SW480 cell line data. Moreover, the two hairpins used to downregulate ESRP1 show variable results- ESRP1 KD-1 negatively affects CD44 whereas ESRP1 KD-2 positively affects CD44.

See above reply to point no. 5 of the “essential revisions” for the clarification of the discrepancy in the data.

C4: Why only exon skipping type of alternatively spliced events were selected for investigation needs justification. Furthermore, in RT-qPCR analyses, GAPDH is used as control. For understanding the relative abundance of specific spliced isoforms of CD44 and NUMB, their expression should also be normalized with Ct value of the constitutive exons of respective genes.

As shown in Author response image 1, exon skipping is by far the most common AS event among the cell lines analyzed. This has justified our focus on this specific alternative splicing modality. As for GAPDH normalization, this choice was justified by our previous analysis of its constant expression levels throughout the samples. However, to satisfy the reviewer’s request, we have now modified by normalizing the CD44 and NUMB isoforms to their constitutive exons. As shown, the results do not change when compared with those obtained by GAPDH normalization (Figure 2, Figure 3, and Figure 3—figure supplement 1).

**Author response image 1. sa2fig1:** 

C5: There is severe inconsistency in the mRNA and protein data in figure 3. As per the western blot analysis of ESRP1 knockdown performed in SW480 cells (figure 3B), CD44 v6 expression is promoted and CD44s is suppressed. Moreover, the knockdown efficiency should also be demonstrated by providing data of ESRP1 bands. In figure 3C and figure 3D, ESRP1 overexpression is leading to upregulation of both NUMB 1/3 and 2/4 isoforms in HCT116 and SW480 cells as per the western blot analysis.

See above reply to point no. 5 of the “essential revisions” for the clarification of the discrepancy in the data. The KD efficiency of the individual shRNA reagents was already reported in the original manuscript in Figure 1E. In the revised manuscript we have better depicted the levels of over-expression and knock down of ESRP1 and ZEB1 (Figure 1E) which also explains, especially in the case of SW480, the variability of the results obtained upon *ESRP1* knockdown when compared with its overexpression. The quantification of the NUMB isoforms bands on western blots by ImageJ was also included (Figure 3A-B and Figure 3—figure supplement 1H).

C6: As the contributions of CD44v6 and CD44s in influencing EMT are vastly different, why HCT116 CD44v6 OE cells are more similar to HCT116 CD44s OE cells (than the parental cells which are naturally more epithelial in nature)? Ref: Page 16 line 8.

In SW480, it appears that overexpression of the CD44s isoform results in a dominant effect on the transcriptome (as observed in the first PC), while overexpression of CD44v6 impacts a smaller component of variation (see second PC). Thus, CD44v6-OE shows more similarity to the parental line. In HCT116, however, this picture is different. The overexpression of both isoforms leads to a considerable transcriptome differences (visible in the first principal component). Nonetheless, pathway analysis still reveals changes in EMT across the isoform groups. Thus, it appears that the isoform-independent effect of CD44 is bigger in HCT116 compared to SW480. In SW480, the effect of EMT is in agreement with the first principal component. In HCT116, this is association is less clear, indicating that additional pathways impact the overexpression of CD44 independent of the particular isoform.